# Reflect-then-Correct: Rebalancing Task Optimization for Generalizable Meta-Reinforcement Learning via Distributional Value Error Reduction

**Min Wang** [1] **Xin Li** [* 1] **Ye He** [1] **Mingzhong Wang** [2] **Yonggang Zhang** [3]

## Abstract

Meta-Reinforcement Learning (Meta-RL) faces significant challenges in non-parametric settings, where vastly different return scales across diverse tasks cause severe gradient interference. Existing categorical solutions attempt to normalize these scales but often fail due to rigid discretization and quantization errors. To address this, we propose Reflect-then-Correct (RTC), a framework that models meta-values using Sinkhorn divergence. By treating distributions as adaptive floating particles, RTC achieves a geometry-aware alignment of distinct meta-task structures. However, while Sinkhorn updates harmonize gradients, they introduce statistical bias via sampling estimation. RTC overcomes this issue by "reflecting" on the temporal accumulation of Bellman inconsistencies through a recursive error model and "correcting" the optimization via adaptive importance weights, which prioritize more accurate transitions for meta-value estimation. We provide theoretical guarantees for this reweighting strategy and demonstrate that RTC outperforms existing baselines on the challenging Meta-World ML-10 and ML-45 benchmarks. Source code is available at https://github.com/MinWangcs/RTC

## 1. Introduction

A fundamental characteristic of human intelligence is the ability to acquire and transfer diverse skills for efficient adaptation to various unforeseen scenarios. Meta-Reinforcement Learning (Meta-RL) (Beck et al., 2023) provides a promising paradigm toward this form of general intelligence by exploiting shared structures across a distribution of training tasks, thereby facilitating rapid adaptation to new tasks.

Context-based meta-RL (Rakelly et al., 2019; Zintgraf et al., 2020), which infers task representations from past experiences, has attracted growing attention for more efficient and stable adaptability. To further improve adaptation to unseen tasks, a branch of research has integrated contrastive learning (Fu et al., 2020), virtual task generation (Lee & Chung, 2021), and bisimulation metrics (Kim et al., 2025) into the inference of task representations, thereby strengthening generalization in out-of-distribution (OOD) tasks. Despite notable advances in improving adaptability, prior meta-RL studies have primarily focused on parametric tasks, where the tasks differ solely in their underlying parameters. For example, in the widely adopted MuJoCo (Todorov et al., 2012), variations in reward functions are commonly induced by modifying the target velocity of a cheetah, with different velocity parameters corresponding to distinct tasks that exhibit insufficient inter-task heterogeneity.

Recent leading meta-RL methods, such as SDVT (Lee et al., 2023) and AMAGO-2 (Grigsby et al., 2024), have shifted attention toward investigating the adaptability of meta-policies in more challenging non-parametric task settings, which require the agent to generalize across semantically diverse tasks with fundamentally different reward structures and physical dynamics. For instance, on the ML-10 protocol of Meta-World (Yu et al., 2019), agents are trained on many diverse manipulation tasks that differ significantly in objectives and dynamics, such as opening doors and playing basketball. They are subsequently evaluated on entirely different unseen tasks, including placing shelves, which exhibit distinct manipulation structures. Despite the non-parametric task setting offering a more applicable and comprehensive evaluation, it presents a unique optimization challenge that is generally circumvented: jointly optimizing tasks with distinct difficulty levels can induce gradient interference, which is further exacerbated in meta-RL due to unknown task identities (Grigsby et al., 2024). Moreover, most prior meta-RL approaches evaluated on distinct meta-tasks commonly neglect the fact that task diversity may introduce imbalanced training losses due to substantial differences in return scales. This oversight often leads to an overemphasis on tasks with large return magnitudes, ultimately causing inefficiency in learning more difficult tasks.

[1]Beijing Institute of Technology, China [2]University of the Sunshine Coast, Australia [3]Jilin University, China. Correspondence to: Xin Li <xinli@bit.edu.cn>.

*Proceedings of the $43^{rd}$ International Conference on Machine Learning*, Seoul, South Korea. PMLR 306, 2026. Copyright 2026 by the author(s).

Distributional RL (Bellemare et al., 2017; Grigsby et al., 2024) provides some relief from this imbalance. By shifting the objective from magnitude-sensitive scalar regression to geometry-aware distribution matching, it prevents tasks with larger return scales from dominating the optimization landscape, thereby fostering more balanced gradient signals. Inspired by this insight, we address the training imbalance problem through the Sinkhorn divergence (Genevay et al., 2018; Feydy et al., 2019). Unlike prior methods (Grigsby et al., 2024) constrained by fixed supports, we treat the meta-value distribution as a collection of floating particles. This formulation enhances meta-task optimization by leveraging Riemannian geometry that aligns returns from different tasks within a unified metric space, enabling more coherent gradient updates. By preserving underlying structural similarities, Sinkhorn divergence enables a shared representation invariant to task-specific return magnitudes, ensuring balanced convergence across the meta-training distribution.

In principle, the ideal optimization objective for meta-value learning should handle two desired criteria: i) adaptively modeling the meta-value distribution with uneven task returns and ii) accurately approximating the meta-value with bootstrapped updates. In this paper, we propose reflect-then-correct (RTC), which learns the uneven meta-value with Sinkhorn updates to address the first criterion and further corrects the estimated value error to rebalance task optimization for the second criterion. Specifically, we start by encoding trajectories to infer task representations, which are concatenated with the state as the input for the meta-value. To balance the training of distinct tasks and become robust to noise, we define an entropy regularized Sinkhorn Temporal Difference (TD) update of the meta-value. The Sinkhorn TD update is prone to statistical bias arising from sampling estimation, which creates a divergence between the empirical loss and the true distance. This bias can distort the meta-value landscape, leading to unstable convergence. To mitigate this problem, we rebalance task optimization via meta-value error correction. By modeling the temporal accumulation of Sinkhorn TD errors recursively, we derive adaptive importance weights that steer the meta-value update toward transitions critical for estimation accuracy. Furthermore, we theoretically prove that weighting training samples based on an error model can reduce the estimation error of the meta-value, thereby enhancing the accuracy of policy learning in complex environments. Our main contributions can be summarized as follows:

1. We propose RTC, which leverages Sinkhorn TD updates to decouple meta-value learning from distinct return scales and defines a recursive error model to prioritize critical transitions, thereby rectifying sampling bias and enhancing estimation accuracy.

2. We provide a theoretical guarantee that meta-value er-

rors can be corrected through reweighting the training distribution based on the estimated Sinkhorn TD error, thereby mitigating error accumulation.

3. We conduct a comprehensive evaluation on the ML-10 and ML-45 benchmarks from Meta-World, which comprise a diverse set of non-parametric tasks (Yu et al., 2019), and the overall results demonstrate the superior adaptability of RTC over baseline methods.

## 2. Related Work

### 2.1. Meta Reinforcement Learning

Meta-RL aims to learn a policy that generalizes across a family of structurally related tasks and enables rapid adaptation to new tasks. To improve the sample inefficiency of gradient-based meta-RL methods (Finn et al., 2017), context-based meta-RL (Rakelly et al., 2019) infers task representations from newly collected trajectories and trains a policy conditioned on these representations to facilitate efficient adaptation. Recent research has mainly concentrated on enhancing the expressiveness of task representations through representation learning methods, such as successor features (Wang et al., 2024), transformers (Xu et al., 2024), and preference queries (Xu & Zhu, 2025). Nonetheless, they are limited to adaptation across minor task variations and exhibit poor robustness to shifts in test-task distributions.

To enhance generalization, a branch of research establishes distinct training and testing distributions, with the goal of improving performance on novel out-of-distribution (OOD) tasks. CCM (Fu et al., 2020) leverages contrastive learning to train a compact context encoder to achieve better generalization across different tasks. Considering the absence of unseen task dynamics during task inference and policy learning, LDM (Lee & Chung, 2021) generates imaginary tasks created from a weighted mixture of the training tasks' latent beliefs. Since LDM focuses solely on reward sample generation, it struggles to handle environments with variations in state transition dynamics. TAVT (Kim et al., 2025) alleviates this problem by leveraging a Bisimulation metric (Ferns & Precup, 2014) to capture task changes in both reward functions and dynamics.

However, the aforementioned approaches typically involve narrowly defined task families, where task diversity arises primarily from minor perturbations of physical parameters within what is effectively a single underlying task. Consequently, when distinct and weakly overlapping trajectories exhibit strong correlations with different task parameters, it becomes difficult to discern whether their context encoder inadvertently exploits spurious, task-irrelevant information, potentially compromising generalization.

In contrast, another line of research, including SDVT (Lee

et al., 2023) and AMAGO-2 (Grigsby et al., 2024), investigates adapting to more distinct task settings. To capture their shared task structures (e.g., both closing curtains and opening drawers involve a pulling action), SDVT decomposes these non-parametric tasks into elementary subtasks and reuses policies learned from common subtasks to enhance adaptation. Similarly, we also employ non-parametric tasks to comprehensively evaluate adaptation efficiency.

## 2.2. Distributional Reinforcement Learning

Distributional RL shifts the focus from estimating expected values to modeling the entire distribution of stochastic returns, thereby capturing the inherent aleatoric uncertainty of the environment. The seminal C51 algorithm (Bellemare et al., 2017) established this paradigm by demonstrating that the categorical projection of the distributional Bellman operator leads to improved stability and performance. Subsequent work (Dabney et al., 2018b;a) extended this framework through quantile-based methods, alleviating the limitations imposed by fixed-support discretization. To better leverage the geometric structure of value distributions, recent advancements (Sun et al., 2024; Kastner et al., 2025) have incorporated optimal transport metrics into the distributional alignment process.

In the context of meta-RL, these distributional perspectives help resolve gradient interference across diverse tasks by providing more informative and structured learning signals. AMAGO-2 (Grigsby et al., 2024) first mitigates this interference by leveraging a categorical representation that converts real-valued targets with varying return scales into classification labels, thereby facilitating improved adaptation and scalability across diverse domains. However, improper discretization of the classification-based value space, especially concerning the number of bins or the chosen value range, can lead to excessive concentration on a subset of bins, resulting in coarse and inaccurate estimates of meta-values. To balance learning across distinct tasks and further improve value estimation accuracy, in this paper, we leverage the Sinkhorn divergence (Feydy et al., 2019) to model the distribution of meta-values and define a TD-based error update to correct meta-value estimation errors.

## 3. Problem Formulation

Meta-RL is defined on a distribution of tasks $p(\mathcal{M})$, where each task is a Markov Decision Process (MDP) (Sutton & Barto, 1998) represented by a tuple $(\mathcal{S}, \mathcal{A}, p, r, \gamma, \rho_0)$, in which $\mathcal{S}$ denotes the state space, $\mathcal{A}$ the action space, $p(s'|s, a)$ the state transition dynamics, $r(s, a, s')$ the reward function, $\rho_0$ the initial state distribution, and $\gamma \in [0, 1)$ the discount factor. In standard RL, the action-value function is defined as $Q^\pi(s, a) := \mathbb{E}[Z^\pi(s, a)] = \mathbb{E}[\sum_{t=0}^{\infty} \gamma^t r(s_t, a_t)]$, $s_{t+1} \sim p(\cdot \mid s_t, a_t)$, which estimates

the expected value of $Z^\pi$. The standard Bellman operator (Bellman, 1966) $\mathcal{F}^\pi$ is defined as $\mathcal{F}^\pi Q(s, a) := \mathbb{E}[r(s, a)] + \gamma \mathbb{E}_{p, \pi}[Q(s', a')]$. In contrast to expectation-based value estimation, distributional RL models the full return distribution. The corresponding distributional Bellman operator (Bellemare et al., 2017) $\mathcal{T}^\pi$ is defined as $\mathcal{T}^\pi Z(s, a) := r(s, a) + \gamma p^\pi Z(s, a)$, where $Z(s, a)$ denotes the value distribution and $p^\pi$ denotes the transition operator.

A task $\mathcal{M}$ is defined as $\{p(s_0), p(s'|s, a), r(s, a, s')\}$ in meta-RL, where different tasks can be obtained by altering dynamics or reward functions. During meta-learning, a meta-policy $\pi(a|s, \mathbf{z})$ is conditioned on the state $s$ and the task representation $\mathbf{z}$. A context encoder is generally leveraged to perform task inference to derive $\mathbf{z}$ based on a set of training tasks sampled from $p(\mathcal{M}_{\text{train}})$. This policy adapts to the present task by considering the history trajectories $\tau$ (referred to as *context*). At test time, the policy should adapt to a new task sampled from $p(\mathcal{M}_{\text{test}})$. To better evaluate the generalization of the meta-trained policy to unseen tasks, consistent with prior works (Kim et al., 2025; Lee et al., 2023), the task sets $\mathbb{M}$ are split into two strictly disjoint training and test sets of MDPs in our setup, i.e., $\mathbb{M} = \mathcal{M}_{\text{train}} \cup \mathcal{M}_{\text{test}}$ and $\mathcal{M}_{\text{train}} \cap \mathcal{M}_{\text{test}} = \emptyset$. Moreover, the meta-policy does not have any prior knowledge about $\mathcal{M}_{\text{test}}$ and cannot interact with $\mathcal{M}_{\text{test}}$ in the meta-training phase. The overall optimization objective of meta-RL can be formulated as follows:

$$
\begin{aligned}
max_\pi &\left\{ \mathcal{R}_\star = \mathbb{E}_{\mathcal{M}_\star \sim p(\mathcal{M})} \left[ \sum_{t=0}^{\infty} \gamma^t r(s_t, a_t) \right] \right\}, \\
&a_t \sim \pi(s_t, \mathbf{z}), \quad \star = \{\text{``train''}, \text{``test''}\}.
\end{aligned} \tag{1}
$$

## 4. Methodology

We propose a framework that integrates Sinkhorn distributional meta-value learning with a recursive error correction mechanism to address non-parametric distinct meta-tasks. First, we leverage the Sinkhorn divergence to represent meta-values as adaptive distributions, yielding a geometry-aware learning signal that aligns diverse task structures. To alleviate the statistical bias and oversmoothing issues arising from Sinkhorn-based sampling, we introduce a recursive error model that captures the temporal accumulation of Bellman inconsistencies. This model produces adaptive importance weights to rebalance task optimization, prioritizing transitions that are most critical for accurate value estimation. Together, these components enable stable convergence and improved generalization across diverse meta-tasks. The overall RTC algorithm is in Algorithm 1.

### 4.1. Sinkhorn Distributional Meta-Value Learning

Given a set of various training tasks drawn from a training task distribution, that is, $\mathcal{M} \sim P(\mathcal{M}_{\text{train}})$, we leverage

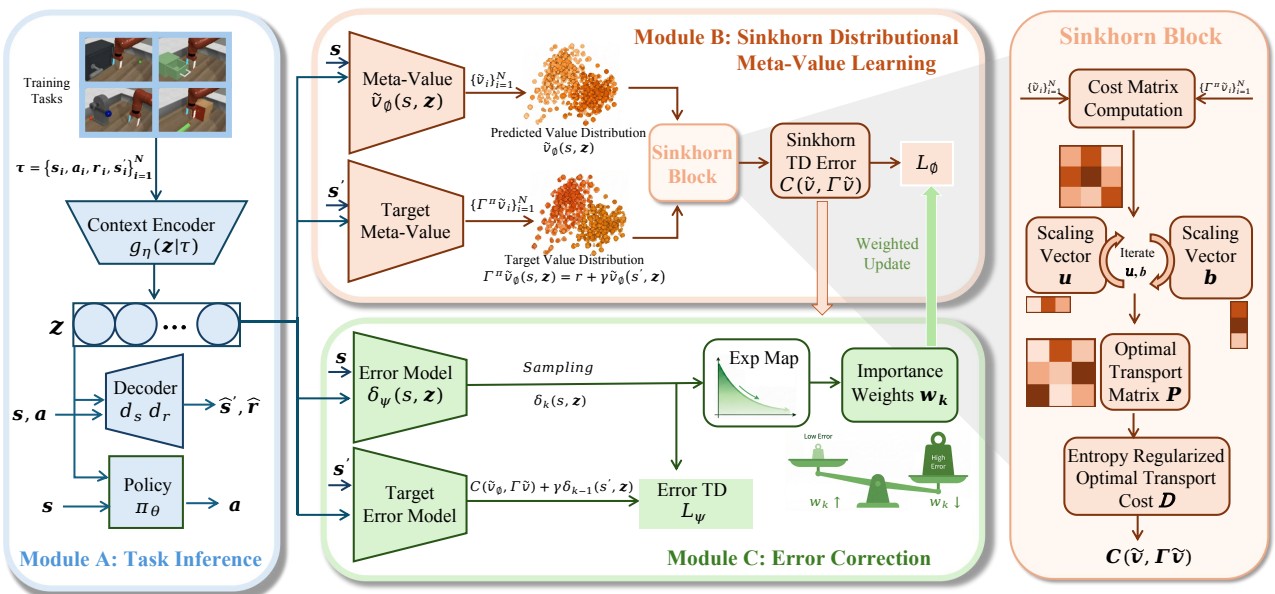

*Figure 1.* Framework of RTC. **Module A (Task Inference)** encodes trajectories to infer task representations **z** for conditioning the policy and meta-value. **Module B (Sinkhorn Distributional Meta-Value Learning)** models meta-values as adaptive distributions and performs an entropy-regularized Sinkhorn TD update to obtain a geometry-aware learning signal that aligns diverse task structures. **Module C (Error Correction)** reflects temporal accumulation of Bellman inconsistencies with a recursive error model and corrects optimization via adaptive importance weights, rebalancing updates toward transitions critical for accurate value estimation.

a context encoder $g$ parameterized by $\eta$ to infer the task representation **z** as follows:

$$\mathbf{z} = g_\eta(\tau_i^{\mathcal{M}}), \tau_i^{\mathcal{M}} = \{s_{i+c}, a_{i+c}, r_{i+c}, s_{i+c+1}\}_{c=0}^H, \quad (2)$$

where $\tau_i^{\mathcal{M}}$ is a trajectory segment of horizon $H$ obtained starting from a randomly chosen $t$ when interacting with the environment composed of $\mathcal{M}$. Similar to variBAD (Zintgraf et al., 2020), we optimize the task inference module with the following objective:

$$\mathbb{E}_{\tau_i}[\mathbb{E}_{g_\eta}[\log p(\tau_{:H}|\mathbf{z}) - KL(g_\eta(\mathbf{z}|\tau_i)||p(\mathbf{z}))]]. \quad (3)$$

The log term corresponds to the reconstruction loss, which is implemented by minimizing the prediction loss of the state decoder $d_s$ and the reward decoder $d_r$, namely $-\mathbb{E}_{(s,a,s',r)\sim\tau_i}[\mathbb{E}_{g_\eta}[\log d_s(\hat{s}'|s,a,\mathbf{z}) + \log d_r(\hat{r}|s,a,\mathbf{z})]]$. The KL term denotes the divergence between the variational posterior $g_\eta$ and the Gaussian prior $p(\mathbf{z})$. Together, they encourage the learned task representations **z** to focus on task-relevant information and to distinguish between different tasks under a unified learning objective.

Following prior works (Bellemare et al., 2017; Dabney et al., 2018b; Chizat et al., 2020), we define the Bellman equation of the meta-value distribution as follows:

$$\Gamma^\pi \tilde{v}(s,\mathbf{z}) := r(s,a,\mathbf{z}) + \gamma\tilde{v}(s',\mathbf{z}), a \sim \pi(\cdot|s,\mathbf{z}), \quad (4)$$

where $\Gamma^\pi$ is a *distributional Bellman operator* and $r(s,a,\mathbf{z})$ is the task-aware reward function. The entropy regularized

optimal transport cost between meta-value $\tilde{v}$ and the TD target $\Gamma^\pi\tilde{v}$ is formulated as:

$$\mathcal{D}(\tilde{v}, \Gamma^\pi\tilde{v}) = \min_{\Pi\in\mathbf{\Pi}(\tilde{v},\Gamma^\pi\tilde{v})} \left\{ \int c(x,y)\,\mathrm{d}\Pi(x,y) + \lambda\mathcal{H}(\Pi, \tilde{v}\otimes\Gamma^\pi\tilde{v}) \right\}, \quad (5)$$

where $\mathbf{\Pi}(\tilde{v},\Gamma^\pi\tilde{v})$ is the set of joint distributions between $\tilde{v}$ and $\Gamma^\pi\tilde{v}$, $c(x,y)$ denotes the cost function, and $\lambda \geq 0$ represents the regularization parameter. The entropy term $\mathcal{H}(\Pi, \tilde{v}\otimes\Gamma^\pi\tilde{v})$ is calculated as $\int(\log\Pi(x,y) - \log\mathrm{d}\tilde{v}(x) - \log\mathrm{d}\Gamma^\pi\tilde{v}(y))\mathrm{d}\Pi(x,y)$.

To compute the integrals, we replace Eq. 5 with the equivalent mini-batch sampling form shown in Eq. 6 (Genevay et al., 2018; Scetbon & Cuturi, 2020). Formally, we define the Sinkhorn TD error of the meta-value as:

$$\mathcal{C}(\tilde{v}, \Gamma^\pi\tilde{v}) = \mathcal{D}(\tilde{v}, \Gamma^\pi\tilde{v}) - \frac{1}{2}\left(\mathcal{D}(\tilde{v},\tilde{v}) + \mathcal{D}(\Gamma^\pi\tilde{v}, \Gamma^\pi\tilde{v})\right),$$

where $\mathcal{D} = \min_{P\in\mathbb{R}_+^{N\times N}}\{\langle P, c\rangle; P\mathbf{1}_N = \mathbf{1}_N, P^\top\mathbf{1}_N = \mathbf{1}_N\}$

$$+ \lambda\sum_{ij}P_{ij}(\log P_{ij} - 1) + \lambda, \quad (6)$$

and $P$ is the matrix form of $\Pi$. Given two sampling sequences $\{\tilde{v}_i\}_{i=1}^N$ and $\{\Gamma^\pi\tilde{v}_j\}_{j=1}^N$, the cost function is calculated as $c_{i,j}(\tilde{v}_i, \Gamma^\pi\tilde{v}_j) = \|\tilde{v}_i - \Gamma^\pi\tilde{v}_j\|^2$.

Due to the $\lambda$-strongly convexity of the entropically regularized objective $\mathcal{D}$, the optimal transport problem admits

a unique minimizer (Genevay et al., 2018; Scetbon & Cuturi, 2020). In particular, the optimal transport plan $P^*$ is expressed in a factored scaling form:

$$\exists u^* \in \mathbb{R}_+^n, b^* \in \mathbb{R}_+^n \text{ s.t. } P^* = \text{diag}\,(u^*)\,\mathcal{K}\,\text{diag}\,(b^*), \tag{7}$$

where $\mathcal{K}_{i,j} = e^{-c_{i,j}/\lambda}$ denotes the Gibbs kernel associated with the cost $c_{ij}$, and $c_{i,j}$ and $\mathcal{K}_{i,j}$ are the $(i,j)$-th elements of the matrices $c$ and $\mathcal{K}$, respectively. This factorized representation reduces the original matrix-valued optimization to the estimation of two scaling vectors, enabling efficient computation through iterative normalization rather than explicit optimization over the full transport matrix. Starting from $b_0 = \mathbf{1}_N$, the Sinkhorn updates alternate between

$$u^{(l+1)} \leftarrow \frac{\mathbf{1}_N}{\mathcal{K}b^{(l)}} \quad \text{and} \quad b^{(l+1)} \leftarrow \frac{\mathbf{1}_N}{\mathcal{K}^\top u^{(l+1)}}, \tag{8}$$

and are guaranteed to converge to the optimal values.

Beyond computational efficiency, this structured solution highlights a key advantage of the Sinkhorn TD error in the meta-task setup. By restricting the transport plan to a smooth, entropically regularized family parameterized by $u$ and $b$, Sinkhorn divergence softly aligns meta-value distributions across different tasks, mitigating gradient interference caused by task heterogeneity while preserving their intrinsic geometric relationships. Consequently, the resulting Sinkhorn TD error is less sensitive to task-specific noise and distributional mismatch, yielding a stable and informative learning signal that naturally promotes knowledge transfer and generalization across distinct tasks.

### 4.2. Rebalance Task Optimization with Meta-Value Error Correction

While Sinkhorn TD updates provide a stable and geometry-aware learning signal, they are susceptible to statistical bias arising from sampling estimation. Furthermore, the entropic regularization used in the Sinkhorn divergence tends to oversmooth distributional discrepancies, often biasing value updates toward dominant tasks or frequently visited states. To address this limitation, we explicitly rebalance the training by adaptively reweighting value updates according to their contribution to the overall value error. This error-aware reweighting aims to correct inaccurate meta-value estimates and improve generalization in the meta-RL setting.

To this end, we formulate the following optimization problem to determine the training distribution $p_k(s, \mathbf{z})$ at each iteration $k$ that minimizes the discrepancy between the learned meta-value $\tilde{v}_k$ and the optimal meta-value $\tilde{v}^*$:

$$\min_{p_k} \mathbb{E}_{d^{\pi_k}} \left[ \mathcal{C}(\tilde{v}_k, \tilde{v}^*) \right] \quad \text{s.t.} \sum_{s, \mathbf{z}} p_k(s, \mathbf{z}) = 1,$$
$$\tilde{v}_k = \arg\min_{\tilde{v}} \mathbb{E}_{p_k} \left[ \mathcal{C}(\tilde{v}, \Gamma^* \tilde{v}_{k-1}) \right], \tag{9}$$

where $d^{\pi_k}$ is the state distribution under policy $\pi$ and $\Gamma^*$ represents the optimal distributional Bellman operator.

To explicitly model the propagation of meta-value estimation errors through Bellman updates, and inspired by (Kumar et al., 2020), we introduce an auxiliary error function $\delta_k$ in a Sinkhorn distributional form, defined via a Bellman-style recursion. While Sinkhorn TD errors capture local Bellman inconsistencies, $\delta_k$ aggregates these discrepancies along the transition dynamics induced by the policy, thereby quantifying how errors in given samples affect future value predictions. Formally, $\delta_k$ is defined recursively as

$$\delta_k(s, \mathbf{z}) = \mathcal{C}(\tilde{v}_k, \Gamma^\pi \tilde{v}_{k-1}) + \gamma \left( \mathcal{P}^{\pi_{k-1}} \delta_{k-1} \right)(s, \mathbf{z}), \tag{10}$$

where $\mathcal{P}^{\pi_{k-1}}$ is the transition matrix under policy $\pi_{k-1}$ and is defined as $\mathcal{P}^\pi \delta(s, \mathbf{z}) := \mathbb{E}_{s' \sim p(\cdot|s,a), a \sim \pi(\cdot|s,\mathbf{z})}[\delta(s', \mathbf{z})]$. We provide an error propagation bound in Lemma 4.1. Then, based on Lemma 4.1, we have the following Theorem 4.2, and the detailed derivation is provided in the Appendix A.

---

**Algorithm 1** RTC algorithm

*# Meta-training Process*
**Input**: training tasks $\mathcal{M}_i^{train}$ from $p(\mathcal{M}_{\text{train}})$, prior Gaussian distribution $p(\mathbf{z})$, replay buffer $\mathcal{B}$, context encoder $g_\eta(\mathbf{z}|\tau)$, state decoder $d_s(\hat{s}'|s, a, \mathbf{z})$, reward decoder $d_r(\hat{r}|s, a, \mathbf{z})$, contextual policy $\pi_\theta(a|s, \mathbf{z})$, meta-value $\tilde{v}_\phi(s, \mathbf{z})$, error model $\delta_\psi(s, \mathbf{z})$

1: **for** each meta-episode **do**
2:    Sample a training task $\mathcal{M}_i^{train} \sim p(\mathcal{M}_{\text{train}})$
3:    Reset replay buffer $\mathcal{B}$
4:    **for** each training step **do**
5:       Add transitions $(s, a, s', r)$ to $\mathcal{B}$
6:       Sample task representations $\mathbf{z} \sim g_\eta(\mathbf{z}|\tau^i)$
7:       **for** each Sinkhorn iteration **do**
8:          Update scaling vectors $u$ and $b$ with Eq. 8
9:       **end for**
10:     Calculate value error $\mathcal{C}(\tilde{v}, \Gamma^\pi \tilde{v})$ with $u$ and $b$
11:    **end for**
12:   Update meta-value $\tilde{v}_\phi$ with Eq. 14
13:   Update error model $\delta_\psi$ with Eq. 15
14:   Update contextual policy $\pi_\theta$ with Eq. 1
15:   Update encoder $g_\eta$, decoders $d_s$ and $d_r$ with Eq. 3
16: **end for**

*# Meta-testing Process*
**Input**: test tasks $\mathcal{M}^{test}$ from $p(\mathcal{M})$

1: Initialize trajectories $\tau^\mathcal{M} = \{\}$
2: **for** m=1, ..., M **do**
3:    Sample task representations $\mathbf{z} \sim g_\eta(\mathbf{z}|\tau^m)$
4:    Roll out policy $\pi_\theta(a|s, \mathbf{z})$ to obtain trajectories $\mathcal{B}_m^\mathcal{M}$
5:    Store the trajectories: $\tau^\mathcal{M} = \tau^\mathcal{M} \cup \mathcal{B}_m^\mathcal{M}$
6: **end for**

---

**Lemma 4.1** (Error Propagation Bound). *For any iteration $k$, the discrepancy between the learned meta-value $\tilde{v}_k$ and the*

*optimal meta-value $\tilde{v}^*$ admits the following upper bound:*

$$\mathcal{C}(\tilde{v}_k, \tilde{v}^*) \leq \mathcal{C}(\tilde{v}_k, \Gamma^* \tilde{v}_{k-1}) + \gamma \mathcal{P}^{\pi_{k-1}} \mathcal{C}(\tilde{v}_{k-1}, \tilde{v}^*)$$
$$+ \frac{2R_{\max}}{1 - \gamma} \max_{s, \mathbf{z}} D_{\mathrm{TV}}(\pi_{k-1}(\cdot \mid s, \mathbf{z}), \pi^*(\cdot \mid s, \mathbf{z})),$$
(11)

*where $\pi_{k-1}$ denotes the iterated policy and $\pi^*$ represents the optimal policy.*

**Theorem 4.2.** *Let $p_k(s, \mathbf{z})$ denote the solution to a relaxed version of the optimization problem in Eq. 9. Then we have*

$$p_k \propto \frac{\exp\left(-\mathcal{C}(\tilde{v}_k, \tilde{v}^*)\right)}{\alpha^*},$$
(12)

*and the corresponding importance weights are given by*

$$w_k(s, \mathbf{z}) = \frac{p_k}{\mu_k} \propto \exp\left(-\frac{\gamma \left[\mathcal{P}^{\pi_{k-1}} \delta_{k-1}\right](s, \mathbf{z})}{\beta}\right), \quad (13)$$

*where $\mu_k$ denotes the sampling distribution, $\alpha^* \in \mathbb{R}^+$ is the Lagrange multiplier with the constraint $\sum_{s, \mathbf{z}} p_k(s, \mathbf{z}) = 1$, and $\beta > 0$ controls the sharpness of reweighting.*

In practice, we perform a weighted Sinkhorn distributional Bellman update for the meta-value network $\tilde{v}_\phi$ by sampling a batch of transitions $\{(s_i, a_i, s_{i+1}, r_i)\}_{i=1}^N \sim \mu$:

$$\mathcal{L}_\phi = \frac{1}{N} \sum_{i=1}^N w_k(s_i, \mathbf{z}) \mathcal{C}(\tilde{v}_\phi(s_i, \mathbf{z}), r_i + \gamma \tilde{v}(s'_i, \mathbf{z})). \quad (14)$$

Meanwhile, the error model $\delta_\psi$ is trained to approximate the recursive error accumulation defined in Eq. 10:

$$\mathcal{L}_\psi = \frac{1}{N} \sum_{i=1}^N \left(\delta_\psi(s_i, \mathbf{z}) - \mathcal{C}(\tilde{v}_\phi, \Gamma \tilde{v}) - \gamma \delta_{\bar{\psi}'}(s'_i, \mathbf{z})\right)^2, \quad (15)$$

where $\delta_{\bar{\psi}'}$ is a target network used to stabilize training and is updated separately, similar to DQN (Mnih et al., 2013). The contextual policy $\pi_\theta$ is updated based on the PPO (Schulman et al., 2017) algorithm.

## 5. Experiments

### 5.1. Experimental Settings

**Meta-World Benchmark**

The Meta-World (Yu et al., 2019) is a well-established benchmark for evaluating meta-RL algorithms. This benchmark consists of 50 qualitatively distinct robotic manipulation tasks, each with 50 parametric variants featuring randomized goals and initial object positions. Two commonly adopted evaluation protocols for evaluating adaptation to heterogeneous tasks are the Meta-World Meta-Learning 10 (ML-10), which includes 10 training tasks and 5 held-out test tasks (indexed 1–10 for training and 11–15 for testing),

and ML-45, which consists of 45 training tasks and 5 test tasks. In this setting, the policy must maximize cumulative reward while simultaneously exploring to infer the unknown task identity over a meta-episode of 5000 steps, comprising 10 rollout episodes, each with a horizon of 500 steps.

**Baselines**

1) **variBAD** (Zintgraf et al., 2020) approximates inference over unknown tasks to derive task representations within a Bayesian RL framework, jointly meta-training a Variational Auto-Encoder (VAE) to reconstruct the next state and reward, along with a meta-policy conditioned on these representations to balance exploration and exploitation when choosing actions under task uncertainty.

2) **LDM** (Lee & Chung, 2021) trains a meta-policy on imaginary tasks to improve generalization to unseen test tasks. The unseen reward dynamics are generated by a decoder that combines mixtures of existing inferred training task representations, with mixture weights sampled randomly from a Dirichlet distribution. The degree of extrapolation of the mixture can be adjusted to adapt to diverse test tasks.

3) **SDVT** (Lee et al., 2023) employs a Gaussian mixture VAE to decompose each task into a collection of shared elementary subtasks. The composition of these subtasks is represented through a latent categorical context, allowing the policy to generalize to new tasks that are comprised of previously encountered subtasks. Similar to LDM, virtual training is also utilized to further enhance generalization to unseen compositions of subtasks.

4) **AMAGO-2** (Grigsby et al., 2024) extends transformer-based meta-RL by reformulating both the actor and critic optimization objectives as classification terms that are invariant to the scale of returns, thereby addressing the issue of imbalanced training losses arising from different return distributions across heterogeneous tasks.

**Evaluation Metrics**

We adhered to the ML-10 and ML-45 evaluation protocols in the Meta-World, which involve few-shot adaptation to 5 new test tasks after training on 10 and 45 meta-training tasks, respectively. All baselines were trained with equal time steps (240M for ML-10 and 390M for ML-45), and subsequent policy evaluations were performed with the same number of time steps. Following SDVT (Lee et al., 2023), a rollout episode is considered successful if the agent succeeds at any timestep, and a meta-episode is deemed successful if its final rollout episode succeeds.

### 5.2. Performance Comparison

Fig. 2 and Table 1 illustrate the testing curves and the converged performance across the more challenging ML-10 and ML-45 of the Meta-World benchmark, respectively. As

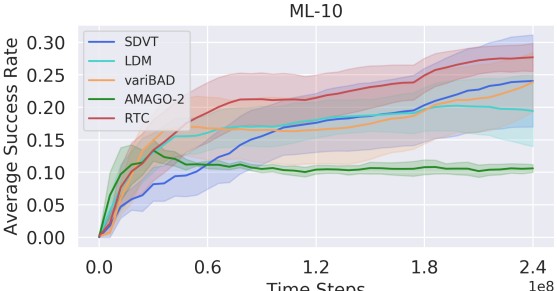 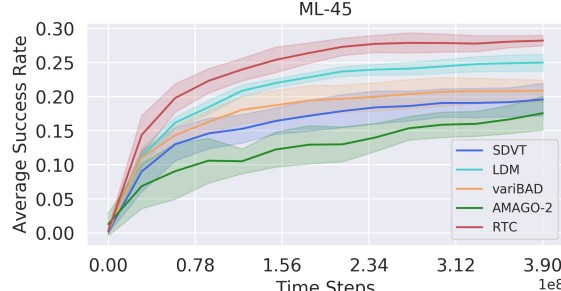

*Figure 2.* Testing average performance on ML-10 and ML-45 over the same 4 random seeds. Our RTC achieves rapid convergence and exhibits superior asymptotic performance compared to baseline methods.

*Table 1.* Converged average train/test success rate ± standard error (%) and test return on ML-10 and ML-45. Bold and underlined entries denote the best and second-best results.

| | **Success Rate (%)** | | | | **Return** | |
| | ML-10 | | ML-45 | | ML-10 | ML-45 |
| Methods | Train | Test | Train | Test | Test | Test |
|---|---|---|---|---|---|---|
| SDVT | $60.61 \pm 11.66$ | $24.16 \pm 11.18$ | $34.69 \pm 9.33$ | $21.19 \pm 2.17$ | $978.93 \pm 170.25$ | $615.40 \pm 61.94$ |
| LDM | $57.34 \pm 10.54$ | $17.92 \pm 11.09$ | $34.41 \pm 2.40$ | $\underline{26.57} \pm 3.67$ | $895.35 \pm 199.53$ | $\underline{712.63} \pm 45.53$ |
| variBAD | $68.95 \pm 10.30$ | $\underline{28.39} \pm 3.76$ | $41.24 \pm 1.40$ | $23.27 \pm 3.57$ | $\underline{1075.50} \pm 207.92$ | $655.70 \pm 42.16$ |
| AMAGO-2 | $\mathbf{98.44} \pm 0.83$ | $10.89 \pm 2.11$ | $\mathbf{79.91} \pm 3.62$ | $24.07 \pm 5.16$ | $841.25 \pm 133.86$ | $549.50 \pm 73.38$ |
| RTC (Ours) | $\underline{79.53} \pm 2.11$ | $\mathbf{30.42} \pm 3.10$ | $\underline{62.62} \pm 1.64$ | $\mathbf{28.66} \pm 2.02$ | $\mathbf{1394.75} \pm 74.93$ | $\mathbf{757.95} \pm 42.82$ |

depicted in Fig. 2, RTC consistently demonstrates excellent adaptation efficiency, surpassing all baseline methods. As shown in Table 1, RTC achieves the highest test success rate and test return with small variance.

Although SDVT achieves strong performance on ML-10, it falls behind LDM and variBAD regarding adaptation efficiency. The potential reason may be that, with a relatively small number of training tasks, as in ML-10, a limited set of subtask categories can adequately capture the shared structural patterns across tasks. However, as the number and diversity of training tasks increase, as in ML-45, representing a broader range of task variations using a limited set of subtasks becomes increasingly challenging. This dilemma may hinder the model's ability to efficiently adapt to new tasks in more diverse environments. Conversely, while LDM outperforms other baselines on ML-45, it lags behind SDVT on ML-10. This discrepancy may be attributed to LDM's virtual training mechanism: when the training tasks are sufficiently diverse and abundant, the decoder can more effectively recombine learned mixtures to generate distinct tasks, thereby enhancing generalization. Surprisingly, variBAD exhibits impressive overall converged performance, indicating its potential to adapt to more distinct tasks by solely reconstructing task structures in the representation space.

In Table 1, AMAGO-2 achieves the highest training success rate among all baselines; however, its final test performance is relatively lower. This discrepancy indicates that the strong training performance of AMAGO-2 does not consistently translate into improved generalization or adaptation at test time. The possible reason is that AMAGO-2 discretizes the value space into categorical bins at an early stage of learning, which may implicitly bias the learned value representation toward the return distribution observed during training. While this design choice effectively stabilizes optimization and alleviates return-scale sensitivity, it may limit the expressiveness of the value representation when test tasks exhibit different return characteristics. Thus, the resulting meta-value estimates become less accurate, potentially affecting adaptation efficiency in unseen task settings.

In contrast, RTC adopts Sinkhorn-based distributional learning for meta-value estimation, substantially reducing reliance on potentially sensitive scalar regression targets and providing a more flexible and informative learning signal across tasks. Moreover, RTC incorporates adaptive importance weighting to refine distributional updates, prioritizing transitions that are most critical for improving estimation accuracy. As a result, RTC consistently demonstrates more rapid adaptation and superior final performance on both training and test tasks across ML-10 and ML-45.

### 5.3. Visualization of Task Representations

To further analyze the structure of the learned task representations in the latent space, we consider the five test task categories in ML-10 and sample 50 task variations per cate-

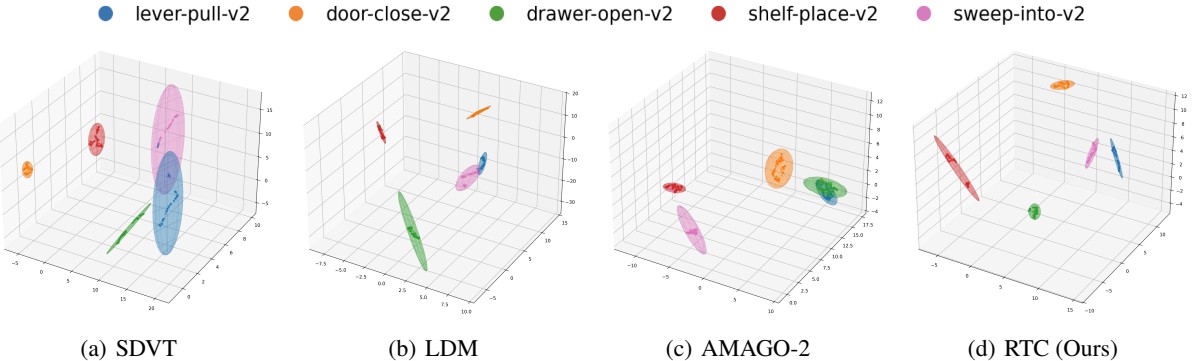

(a) SDVT      (b) LDM      (c) AMAGO-2      (d) RTC (Ours)

*Figure 3.* Three-dimensional projection of task representations related to different tasks on the ML-10 benchmark.

gory. For each variation, we collect context segments from all evaluation episodes and compute the mean task representation. This yields one representative embedding vector per variation, resulting in 250 embeddings in total.

We apply the UMAP method (McInnes & Healy, 2018) to map these embeddings into a 3D space for visualization. To characterize the intra-category variability after projection, we compute the empirical mean and covariance of the 3D projected points for each category. We then visualize the estimated Gaussian distribution as an ellipsoid in the 3D space. Fig. 3 presents the resulting latent-space geometry across different methods. Our RTC exhibits clearly separated clusters for the five task categories, with ellipsoids showing limited overlap. This indicates that RTC learns task representations that are both discriminative across tasks and consistent within each category. In contrast, the baseline methods show noticeable overlap among ellipsoids.

### 5.4. Validation of Task Representation for Learning Shared Structures

We further validate the role of task representation by visualizing the mean task representation vector for each environment variant as a heatmap. As shown in Fig. 4, some tasks consistently exhibit large values along specific latent dimensions, effectively dominating those axes. For instance, pick-place-v2 and shelf-place-v2 show similar activation patterns and peak at X4 and X5. Similarly, drawer-open-v2 and window-open-v2 exhibit strong and consistent activations at X5. This axis-aware specialization suggests that semantically similar tasks share a subset of dominant latent dimensions, yielding a more structured and interpretable latent space with task entanglement.

### 5.5. Ablation Studies

We conduct ablation studies to evaluate the contributions of Sinkhorn-based distributional meta-value learning and the meta-value error correction module. As illustrated in

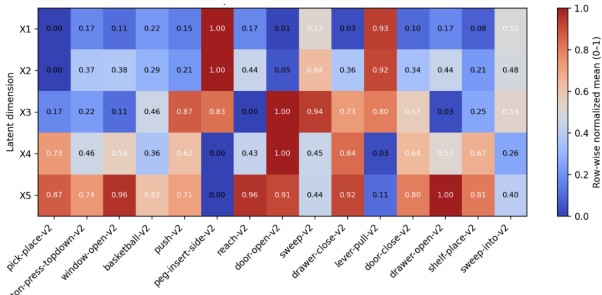

*Figure 4.* Heatmap of mean task representation vectors inferred by RTC on ML-10. This suggests that our RTC captures similar representations for tasks that are structurally similar.

Fig. 5, a consistent performance degradation is observed on both ML-10 and ML-45 when the error correction module is ablated. Their performance further declines upon the ablation of the Sinkhorn distributional value learning module, indicating the indispensable role of both components.

The ablation results demonstrate that both modules are essential for addressing the unique challenges of meta-RL when adapting to distinct tasks. Specifically, the Sinkhorn TD update of the meta-value is crucial for geometry-aware alignment, enabling the soft matching of value distributions across tasks and balancing task optimization effects caused by varying return scales. However, Sinkhorn TD updates alone are insufficient due to sampling-induced statistical bias and the oversmoothing effects of entropic regularization. The error correction module explicitly models the temporal accumulation of Bellman errors and applies adaptive importance weights to rebalance the optimization process. This ensures that the agent prioritizes transitions critical for accuracy rather than being biased toward dominant tasks. The core synergy between the two modules lies in their complementary roles: Sinkhorn alignment establishes structured cross-task value geometry, while error correction adaptively regulates temporal learning dynamics over time. Together, they effectively enable robust generalization and rapid adaptation to more complex meta-tasks.

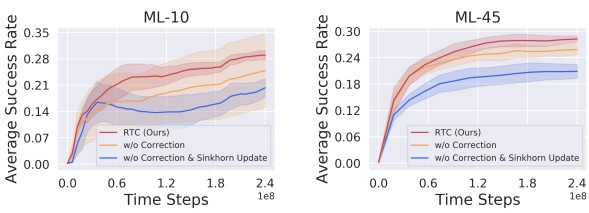

*Figure 5.* Ablation study.

## 5.6. Visualization of the Role of Error Correction

To analyze the optimization behavior, as illustrated in Fig. 6, we visualize the local loss geometry of the meta-value network by plotting the loss surface around the trained parameters on ML-45. In Fig. 6(a), the meta-value network lacking error correction exhibits a sharp and rugged landscape, indicating high sensitivity to perturbations and poor generalization. In contrast, in Fig. 6(b), the landscape incorporating value error correction is notably smoother and flatter, suggesting enhanced robustness across distinct tasks. This observation aligns with the role of adaptive importance weights, which stabilize convergence and improve generalization by rebalancing task updates, mitigating sampling bias, and suppressing error accumulation.

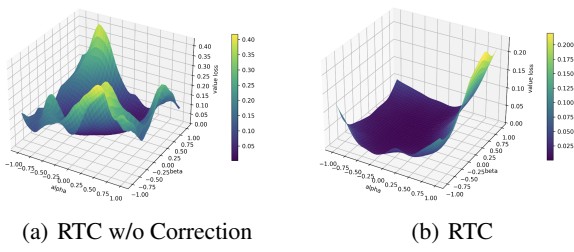

(a) RTC w/o Correction          (b) RTC

*Figure 6.* Landscape of the meta-value network on ML-45.

## 6. Conclusion

Our work addresses the training imbalance problem in meta-RL arising from highly distinct tasks. We propose Reflect-then-Correct (RTC), which leverages the Sinkhorn divergence to achieve geometry-aware alignment across diverse tasks while explicitly correcting statistical bias. By modeling the temporal accumulation of Bellman inconsistencies through a recursive error model and reweighting transitions accordingly, RTC combines flexible distributional representations with error-aware optimization. Our theoretical analysis establishes error reduction guarantees for this correction strategy, and empirical results on ML-10 and ML-45 of the Meta-World benchmark demonstrate substantially improved adaptability to semantically diverse tasks.

## Acknowledgments

The authors would like to thank the anonymous reviewers for their valuable feedback. This work was partially supported by the National Natural Science Foundation of China (Grant No. 62276024), the National Key R&D Program of China (Grant No. 2025YFC3309100), the Beijing Natural Science Foundation (Grant No. 4262066), the Fundamental Research Funds for the Central Universities, Jilin University (Grant No. 93K172025K01), and the Fundamental Research Funds for the Central Universities (Grant No. 2025CX01010).

## Impact Statement

Our proposed RTC method for meta-reinforcement learning with more distinct tasks has the potential to positively impact various domains significantly. By enabling more efficient and adaptable learning in complex and distinct tasks, RTC could accelerate progress in various fields. In robotics and automation, the ability of RTC to handle distinct tasks could lead to more versatile and resilient systems, potentially improving manufacturing efficiency. However, these advancements come with the potential risk of perpetuating or amplifying biases if the training data or task distributions are not representative. Furthermore, the deployment of such advanced AI systems, especially in critical applications, requires careful consideration of safety, robustness, and potential dual-use concerns.

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

# A. Proof

**Lemma A.1** (Error Propagation Bound). *For any iteration $k$, the discrepancy between the learned meta-value $\tilde{v}_k$ and the optimal meta-value $\tilde{v}^*$ admits the following upper bound:*

$$\mathcal{C}(\tilde{v}_k, \tilde{v}^*) \leq \mathcal{C}(\tilde{v}_k, \Gamma^* \tilde{v}_{k-1}) + \gamma \mathcal{P}^{\pi_{k-1}} \mathcal{C}(\tilde{v}_{k-1}, \tilde{v}^*) + \frac{2R_{\max}}{1-\gamma} \max_{s,\mathbf{z}} D_{\mathrm{TV}}(\pi_{k-1}(\cdot \mid s, \mathbf{z}), \pi^*(\cdot \mid s, \mathbf{z})), \tag{16}$$

*where $\pi_{k-1}$ denotes the iterated policy and $\pi^*$ is the optimal policy.*

*Proof.*

$$
\begin{aligned}
\mathcal{C}(\tilde{v}_k, \tilde{v}^*) &\overset{(a)}{\leq} \mathcal{C}(\tilde{v}_k, \Gamma^* \tilde{v}_{k-1}) + \mathcal{C}(\Gamma^* \tilde{v}_{k-1}, \Gamma^* \tilde{v}^*) \\
&\overset{(b)}{\leq} \mathcal{C}(\tilde{v}_k, \Gamma^* \tilde{v}_{k-1}) + \gamma \mathcal{C}(\mathcal{P}^{\pi_{k-1}} \tilde{v}_{k-1}, \mathcal{P}^{\pi^*} \tilde{v}^*) \\
&\overset{(c)}{\leq} \mathcal{C}(\tilde{v}_k, \Gamma^* \tilde{v}_{k-1}) + \gamma \mathcal{C}\left(\mathcal{P}^{\pi_{k-1}} \tilde{v}_{k-1}, \mathcal{P}^{\pi_{k-1}} \tilde{v}^*\right) + \gamma \mathcal{C}\left(\mathcal{P}^{\pi_{k-1}} \tilde{v}^*, \mathcal{P}^{\pi^*} \tilde{v}^*\right) \\
&\overset{(e)}{\leq} \mathcal{C}(\tilde{v}_k, \Gamma^* \tilde{v}_{k-1}) + \gamma \mathcal{P}^{\pi_{k-1}} \mathcal{C}\left(\tilde{v}_{k-1}, \tilde{v}^*\right) + \frac{2R_{\max}}{1-\gamma} \max_{s,\mathbf{z}} D_{\mathrm{TV}}(\pi_{k-1}, \pi^*),
\end{aligned} \tag{17}
$$

where (a) applies the triangle inequality of Sinkhorn divergence (Genevay et al., 2018), decomposing the total error into the current Bellman error and the propagated error from the previous iteration. Step (b) uses the $\gamma$-contraction property of the optimal Bellman operator $\Gamma^*$ (Sun et al., 2024) to relate the propagated error to the previous iteration. Step (c) applies the triangle inequality, separating the error into value propagation and policy mismatch contributions. Finally, step (e) bounds the policy mismatch term using the standard total variation inequality together with the reward bound $R_{\max}$. □

**Theorem A.2.** *Let the training distribution $p_k(s, \mathbf{z})$ at each iteration $k$ denote the solution to a relaxed version of the optimization problem that minimizes the discrepancy between the learned meta-value $\tilde{v}_k$ and the optimal meta-value $\tilde{v}^*$:*

$$\min_{p_k} \mathbb{E}_{d^{\pi_k}}\left[\mathcal{C}(\tilde{v}_k, \tilde{v}^*)\right] \quad s.t. \sum_{s,\mathbf{z}} p_k(s, \mathbf{z}) = 1, \quad \tilde{v}_k = \arg\min_{\tilde{v}} \mathbb{E}_{p_k}\left[\mathcal{C}(\tilde{v}, \Gamma^* \tilde{v}_{k-1})\right], \tag{18}$$

*where $d^{\pi_k}$ is the state distribution under policy $\pi$, and $\Gamma^*$ represents the optimal distributional Bellman operator.*

*Then we have*

$$p_k \propto \frac{\exp\left(-\mathcal{C}(\tilde{v}_k, \tilde{v}^*)\right)}{\alpha^*}, \tag{19}$$

*and the corresponding importance weights are given by*

$$w_k(s, \mathbf{z}) = \frac{p_k}{\mu_k} \propto \exp\left(-\frac{\gamma \left[\mathcal{P}^{\pi_{k-1}} \delta_{k-1}\right](s, \mathbf{z})}{\beta}\right), \tag{20}$$

*where $\mu_k$ denotes the sampling distribution, $\alpha^* \in \mathbb{R}^+$ is the Lagrange multiplier with the constraint $\sum_{s,\mathbf{z}} p_k(s, \mathbf{z}) = 1$, and $\beta > 0$ controls the sharpness of reweighting.*

*Proof.* Following prior work (Kumar et al., 2020), we reformulate the primal optimization problem in Eq. 18 into a more tractable form using the Fenchel-Young inequality (Rockafellar, 1997), while preserving the same optimal solution. By adopting a *soft-min* convex objective, the problem is reformulated as:

$$\min_{p_k} -\log\left(\sum_{s,\mathbf{z}} \exp\left(-\mathcal{C}(\tilde{v}_k, \tilde{v}^*)\right)\right) \quad s.t. \sum_{s,\mathbf{z}} p_k(s, \mathbf{z}) = 1, \quad \tilde{v}_k = \arg\min_{\tilde{v}} \mathbb{E}_{p_k}\left[\mathcal{C}(\tilde{v}, \Gamma^* \tilde{v}_{k-1})\right]. \tag{21}$$

The corresponding Lagrangian functional for this optimization is given by:

$$\mathcal{J}(p_k; \alpha, \omega) = -\log\left(\sum_{s,\mathbf{z}} \exp\left(-\mathcal{C}(\tilde{v}_k, \tilde{v}^*)\right)\right) + \alpha\left(\sum_{s,\mathbf{z}} p_k(s, \mathbf{z}) - 1\right) - \omega^T p_k, \tag{22}$$

where $\alpha \in \mathbb{R}$ and $\omega \in \mathbb{R}_+^n$ denote the Lagrange multipliers associated with the equality and non-negativity constraints, respectively. By setting the gradient of $\mathcal{J}$ with respect to $p_k$ to zero, we can approximately obtain an optimal $p_k$ as follows:

$$p_k \propto \frac{\exp\left(-\mathcal{C}(\tilde{v}_k, \tilde{v}^*)\right)}{\alpha^*}. \tag{23}$$

We define the estimated meta-value error $\delta_k$ as follows:

$$\delta_k = \sum_{i=1}^{k} \gamma^{k-i} \left(\prod_{j=i}^{k-1} \mathcal{P}^{\pi_j}\right) \mathcal{C}(\tilde{v}_i, \Gamma^* \tilde{v}_{i-1}), \tag{24}$$

which admits the recursive form: $\delta_k(s, \mathbf{z}) = \mathcal{C}(\tilde{v}_k, \Gamma^\pi \tilde{v}_{k-1}) + \gamma \left(\mathcal{P}^{\pi_{k-1}} \delta_{k-1}\right)(s, \mathbf{z})$.

By recursively expanding the error propagation bound established in Lemma A.1, we decompose the total meta-value error $\mathcal{C}(\tilde{v}_k, \tilde{v}^*)$ into three distinct components:

$$
\begin{aligned}
\mathcal{C}\left(\tilde{v}_k, \tilde{v}^*\right) \leq & \underbrace{\sum_{i=1}^{k} \gamma^{k-i} \left(\prod_{j=i}^{k-1} \mathcal{P}^{\pi_j}\right) \mathcal{C}\left(\tilde{v}_i, \Gamma^* \tilde{v}_{i-1}\right)}_{\text{(i) Accumulated Bellman Error } (\delta_k)} + \underbrace{\gamma^k \left(\prod_{j=0}^{k-1} \mathcal{P}^{\pi_j}\right) \mathcal{C}\left(\tilde{v}_0, \tilde{v}^*\right)}_{\text{(ii) Initial Bias}} \\
& + \underbrace{\sum_{i=1}^{k} \gamma^{k-i} \left(\prod_{j=i}^{k-1} \mathcal{P}^{\pi_j}\right) \frac{2R_{\max}}{1-\gamma} \max_{s, \mathbf{z}} D_{\text{TV}}(\pi_{i-1}, \pi^*)}_{\text{(iii) Accumulated Policy Mismatch}}.
\end{aligned} \tag{25}
$$

In the above decomposition, term (i) represents the cumulative meta-value approximation errors during the Bellman update across $k$ iterations, as defined in Eq. 24. Term (ii) accounts for the initial estimation bias, which decays geometrically at rate $\gamma^k$, and term (iii) captures the policy mismatch error arising from the discrepancy between the iterated policy $\pi_{i-1}$ and the optimal policy $\pi^*$. For subsequent analysis, we bound the latter two terms by constants $c_1$ and $c_2$, respectively, yielding the compact upper bound: $\mathcal{C}(\tilde{v}_k, \tilde{v}^*) \leq \delta_k + c_1 + c_2$.

Since it is difficult to directly obtain samples from the training distribution $p_k$ via online interaction, we use samples from a buffer distribution $\mu_k$ and reweight these samples using importance sampling. Substituting the above upper bound back into the expression for the optimal weights $w_k$, we derive the following formulation of the sample weights:

$$w_k = \frac{p_k}{\mu_k} \propto \exp\left(\frac{-c_2 - c_1 - \gamma \left[\mathcal{P}^{\pi_{k-1}} \delta_{k-1}\right](s, \mathbf{z})}{\alpha^*}\right).$$

Importantly, employing this expression for $w_k$ serves to down-weight transitions associated with high error accumulation, thereby inducing a "conservative" distribution that prioritizes reliable samples. By subsuming the constants $c_1, c_2$, and $\alpha^*$ into a single temperature parameter $\beta$, we arrive at the practical weighting rule:

$$w_k(s, \mathbf{z}) \propto \exp\left(-\frac{\gamma \left[\mathcal{P}^{\pi_{k-1}} \delta_{k-1}\right](s, \mathbf{z})}{\beta}\right). \tag{26}$$

$\square$

## B. Experimental Settings

### B.1. Meta-World Benchmark

Meta-World is an open-source simulated benchmark for evaluating multi-task reinforcement learning and meta-reinforcement learning (meta-RL) on diverse robotic manipulation problems. It comprises 50 qualitatively distinct manipulation tasks, each further incorporating parametric variations (e.g., randomized object and goal positions) to promote generalization beyond fixed configurations. In our experiments, we follow the widely adopted Meta-World meta-learning protocols and evaluate RTC and the baselines on ML-10 and ML-45, both of which assess few-shot adaptation to held-out test tasks after meta-training on a predefined set of training tasks.

**Environment interface.** All tasks are performed by a simulated Sawyer robot. The action space corresponds to the 3D end-effector position, while the observation space consists of Cartesian positions of the end-effector, object(s), and the goal or a second object. For evaluation, Meta-World additionally defines an interpretable success metric based on whether the task-relevant object reaches the target goal.

**Meta-learning protocols (ML-10 / ML-45).** Following the Meta-World evaluation design for generalization to new tasks, 5 tasks are held out for meta-testing, and policies are meta-trained on either 10 tasks (ML-10) or 45 tasks (ML-45). The protocols randomize object and goal positions. Each meta-episode spans 5000 environment steps, composed of 10 rollout episodes with a horizon of 500 steps each, during which the policy must maximize return while exploring to infer the unknown task identity. Figure 7 illustrates the 45 meta-training tasks and the 5 meta-test tasks in Meta-World ML-45, and Table 2 lists the detailed task descriptions for the 10 meta-training tasks and the meta-test tasks in ML-10.

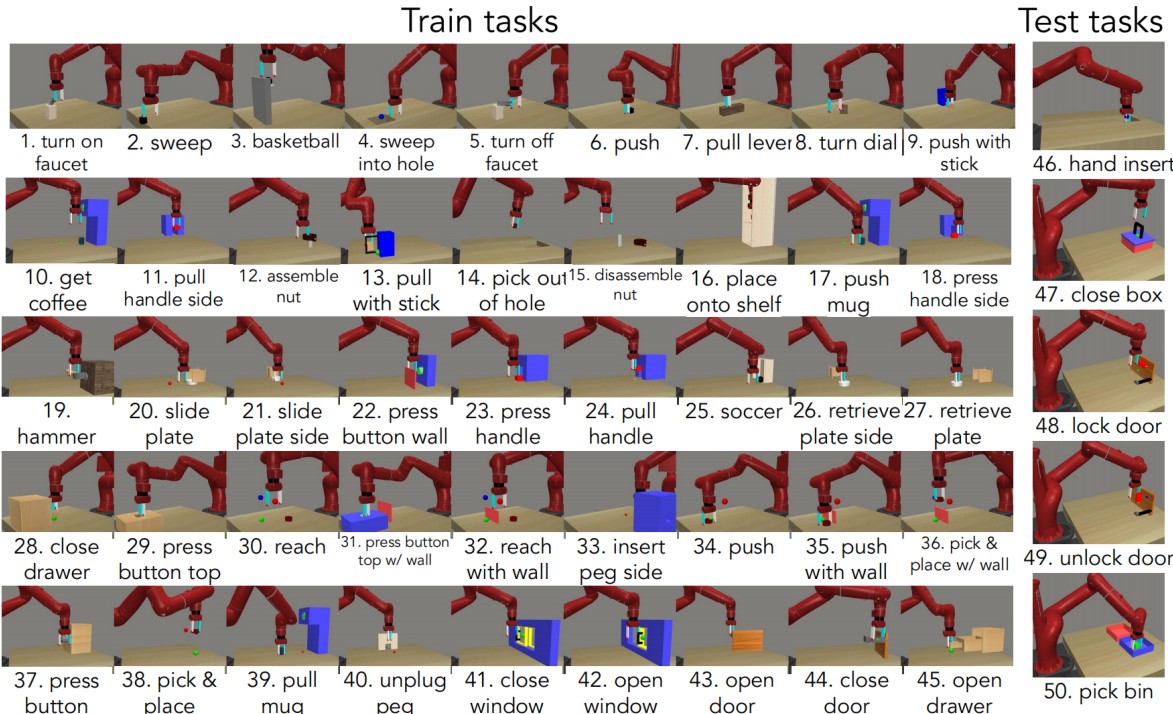

*Figure 7.* Tasks in Meta-World. Image from (Yu et al., 2019)

**Training steps and evaluation metric.** We train all methods using the same number of interaction steps: 240M steps for ML-10 and 390M steps for ML-45. Following the evaluation protocol adopted in prior study (Yu et al., 2019), a rollout episode is considered successful if the agent succeeds at any timestep, while a meta-episode is considered successful if its final rollout episode succeeds. Success rates are averaged across four random seeds. Following SDVT, we use 50 trials per test task, each with a different goal position.

### B.2. Hyperparameters

The hyperparameters of RTC are outlined in Table 3.

### B.3. Network Architecture

As summarized in Table 4, we use multilayer perceptrons for the state embedding, the reward embedding, the action embedding, the policy network, and the reward and state decoders, together with a GRU-based recurrent encoder for task inference. Except for AMAGO-2 (Grigsby et al., 2024), which uses a transformer architecture, the architectures and layer widths are kept consistent across RTC and baselines following SDVT (Lee et al., 2023) to ensure a fair comparison.

*Table 2.* Meta-World ML-10 task descriptions

| Task | Description |
|---|---|
| **ML10 Meta-training tasks (10)** | |
| pick&place | Pick and place a puck to a goal. Randomize puck and goal positions. |
| press button top | Press a button from the top. Randomize button positions. |
| open window | Push and open a window. Randomize window positions. |
| basketball | Dunk the basketball into the basket. Randomize basketball and basket positions. |
| push | Push the puck to a goal. Randomize puck and goal positions. |
| insert peg side | Insert a peg sideways. Randomize peg and goal positions. |
| Reach | Reach a goal position. Randomize the goal positions. |
| open door | Open a door with a revolving joint. Randomize door positions. |
| sweep | Sweep a puck off the table. Randomize puck positions. |
| close drawer | Push and close a drawer. Randomize the drawer positions. |
| **ML10 Meta-test tasks (5)** | |
| pull lever | Pull a lever down 90 degrees. Randomize lever positions. |
| close door | Close a door with a revolvinig joint. Randomize door positions. |
| open drawer | Open a drawer. Randomize drawer positions. |
| place onto shelf | pick and place a puck onto a shelf. Randomize puck and shelf positions. |
| sweep into hole | Sweep a puck into a hole. Randomize puck positions. |

*Table 3.* Hyperparameters

| Description | Value |
|---|---|
| Rollout episode horizon | 500 |
| Max rollouts per task | 10 |
| Discount factor for rewards | 0.99 |
| Number of parallel processes | 6 |
| Number of frames to train | 24e7(for ML-10), 39e7(for ML-45) |
| Learning rate of task inference | 0.001 |
| Number of steps for task inference | 100 |
| Buffer size of task inference | 1000 |
| Update steps of task inference per meta-iteration | 10 |
| Dimensionality of task representation | 5 |
| Optimiser | Adam |
| Learning rate of policy | 7e-4 |
| Optimizer epsilon of policy | 1e-8 |
| PPO update epochs | 5 |
| Number of steps per policy update | 5000 |
| Minimum standard deviation | 0.5 |
| Maximum standard deviation | 1.5 |
| Entropy term coefficient | 0.001 |

*Table 4.* Network architecture.

| Network | Architecture | Layers |
|---|---|---|
| State embedding | MLP | [32] |
| Reward embedding | MLP | [16] |
| Action embedding | MLP | [16] |
| Recurrent encoder | GRU | [256] |
| Reward decoder | MLP | [64, 64, 32, 1] |
| Transition decoder | MLP | [64, 64, 32, 40] |
| Policy | MLP | [256, 256, 4] |

## C. Detailed Experiment Result

In this section, we report detailed per-task performance on Meta-World. Specifically, Table 5 summarizes the per-task test returns on ML-10, while Table 6 reports the per-task test returns on ML-45. These results complement the overall performance reported in the main paper by providing a task-level perspective on the performance and generalization capability of our method.

*Table 5.* Per-task average return (%) on ML-10 (mean $\pm$ standard error).

| Task | RTC | LDM | SDVT | VariBAD |
|---|---|---|---|---|
| pick-place-v2 | $3612.0 \pm \mathbf{63.1}$ | $\mathbf{3628.0} \pm 101.5$ | $3434.4 \pm 118.2$ | $3428.5 \pm 263.5$ |
| button-press-topdown-v2 | $\mathbf{3944.5} \pm \mathbf{67.7}$ | $2009.2 \pm 1463.7$ | $2406.0 \pm 1719.7$ | $3462.7 \pm 657.5$ |
| window-open-v2 | $\mathbf{2425.9} \pm 327.3$ | $1381.9 \pm 1050.1$ | $1384.2 \pm 982.0$ | $2105.1 \pm \mathbf{200.8}$ |
| basketball-v2 | $\mathbf{4464.6} \pm 91.3$ | $3814.9 \pm 632.1$ | $4249.0 \pm \mathbf{81.1}$ | $4297.2 \pm 180.2$ |
| push-v2 | $4827.4 \pm 31.8$ | $4790.7 \pm 84.7$ | $4818.6 \pm 31.9$ | $\mathbf{4837.0} \pm \mathbf{27.4}$ |
| peg-insert-side-v2 | $\mathbf{3278.3} \pm \mathbf{93.2}$ | $2706.0 \pm 154.1$ | $3134.1 \pm 122.6$ | $3122.8 \pm 275.7$ |
| reach-v2 | $\mathbf{2531.5} \pm \mathbf{550.3}$ | $1172.6 \pm 823.3$ | $1074.6 \pm 757.9$ | $1389.2 \pm 570.1$ |
| door-open-v2 | $\mathbf{4463.8} \pm 64.2$ | $4208.3 \pm 87.2$ | $4137.5 \pm \mathbf{33.0}$ | $4427.5 \pm 102.4$ |
| sweep-v2 | $\mathbf{3899.6} \pm \mathbf{163.6}$ | $1743.4 \pm 1523.9$ | $1600.4 \pm 1404.2$ | $3292.4 \pm 1119.2$ |
| drawer-close-v2 | $2390.1 \pm 1027.8$ | $1528.8 \pm 1107.6$ | $2213.7 \pm 1651.9$ | $\mathbf{3006.1} \pm \mathbf{574.5}$ |
| lever-pull-v2 | $2299.1 \pm 278.0$ | $2043.3 \pm \mathbf{120.0}$ | $2430.5 \pm 135.8$ | $\mathbf{2601.2} \pm 176.4$ |
| door-close-v2 | $230.1 \pm \mathbf{54.9}$ | $152.1 \pm 68.9$ | $832.6 \pm 560.7$ | $\mathbf{1477.6} \pm 908.1$ |
| drawer-open-v2 | $\mathbf{417.8} \pm \mathbf{177.3}$ | $296.3 \pm 226.1$ | $308.6 \pm 221.4$ | $410.9 \pm 183.2$ |
| shelf-place-v2 | $\mathbf{2340.8} \pm \mathbf{449.3}$ | $739.0 \pm 506.6$ | $1073.0 \pm 850.1$ | $1374.2 \pm 660.8$ |
| sweep-into-v2 | $235.0 \pm 22.2$ | $\mathbf{316.4} \pm 20.8$ | $307.5 \pm \mathbf{15.8}$ | $314.5 \pm 36.8$ |

*Table 6.* Per-task average return (%) on ML-45 (mean ± standard error).

| Task | RTC | SDVT | LDM | VariBAD |
|---|---|---|---|---|
| handle-pull-side-v2 | **2437.5** ± **100.2** | 937.6 ± 757.2 | 877.9 ± 315.3 | 1547.8 ± 279.8 |
| door-close-v2 | **1939.2** ± 387.1 | 724.1 ± 967.0 | 232.6 ± **210.3** | 517.1 ± 272.9 |
| disassemble-v2 | **3233.3** ± **403.4** | 2014.1 ± 964.3 | 1661.9 ± 646.0 | 2631.7 ± 733.3 |
| window-open-v2 | **3242.3** ± **378.6** | 2048.3 ± 978.5 | 1584.4 ± 512.3 | 2570.7 ± 738.1 |
| sweep-into-v2 | 2668.6 ± **205.6** | 2193.9 ± 364.3 | **2672.3** ± 421.3 | 2388.1 ± 320.4 |
| dial-turn-v2 | 3087.2 ± **135.9** | **3110.4** ± 349.5 | 2494.8 ± 293.3 | 2553.0 ± 386.6 |
| plate-slide-back-side-v2 | **2194.3** ± 420.7 | 227.2 ± **31.4** | 452.6 ± 137.4 | 913.8 ± 552.1 |
| plate-slide-back-v2 | **849.2** ± 165.7 | 204.8 ± 212.4 | 194.2 ± **41.3** | 211.8 ± 86.0 |
| handle-pull-v2 | **1000.9** ± 329.3 | 364.7 ± 153.8 | 431.4 ± **89.0** | 427.2 ± 164.7 |
| pick-place-wall-v2 | **1731.2** ± 655.0 | 1535.7 ± 593.0 | 1488.4 ± **269.1** | 1542.6 ± 352.0 |
| button-press-wall-v2 | **3449.5** ± 202.7 | 1230.4 ± 1591.4 | 191.7 ± **28.4** | 210.5 ± 121.0 |
| soccer-v2 | **4336.7** ± **55.6** | 4166.0 ± 288.4 | 3939.5 ± 323.7 | 4276.2 ± 90.0 |
| push-back-v2 | **4277.6** ± 137.0 | 1819.5 ± 893.8 | 1736.9 ± **73.2** | 1833.5 ± 657.0 |
| coffee-push-v2 | **4638.1** ± **17.6** | 3798.1 ± 756.7 | 4347.0 ± 164.8 | 4285.7 ± 122.2 |
| sweep-v2 | **3944.0** ± 466.8 | 2306.8 ± 572.9 | 2519.8 ± 317.9 | 3126.2 ± **283.5** |
| push-v2 | **4528.8** ± 128.4 | 3184.0 ± 994.6 | 4278.5 ± **83.9** | 4301.4 ± 88.4 |
| handle-press-v2 | **3866.9** ± 821.7 | 3509.3 ± 514.2 | 2792.0 ± **128.2** | 3453.0 ± 194.4 |
| peg-unplug-side-v2 | 429.6 ± 67.5 | 470.8 ± 10.0 | 470.1 ± 16.4 | **488.9** ± **8.5** |
| plate-slide-side-v2 | **4680.0** ± 98.4 | 4167.5 ± 395.5 | 4010.7 ± 551.3 | 4651.5 ± **79.7** |
| faucet-close-v2 | **4603.8** ± 127.8 | 3941.8 ± 1094.5 | 4323.6 ± **79.3** | 4549.7 ± 225.4 |
| button-press-topdown-wall-v2 | **1185.4** ± 627.0 | 35.5 ± **1.9** | 140.0 ± 156.6 | 71.5 ± 50.2 |
| push-wall-v2 | **1711.3** ± **37.8** | 923.4 ± 240.0 | 1407.6 ± 63.9 | 1362.1 ± 221.0 |
| stick-push-v2 | **496.0** ± **50.8** | 361.4 ± 61.7 | 385.9 ± 71.1 | 415.9 ± 57.3 |
| basketball-v2 | **1052.0** ± 183.1 | 358.6 ± 250.6 | 244.2 ± 93.2 | 410.2 ± **77.5** |
| faucet-open-v2 | **2608.8** ± 481.2 | 804.1 ± 971.0 | 318.1 ± **165.1** | 1256.9 ± 762.9 |
| hammer-v2 | **918.7** ± 796.2 | 125.5 ± 141.7 | 22.0 ± **7.0** | 39.6 ± 17.8 |
| peg-insert-side-v2 | **3029.1** ± 155.3 | 1492.4 ± 105.6 | 1710.4 ± **76.1** | 2143.6 ± 137.1 |
| drawer-open-v2 | **1440.6** ± 276.5 | 188.5 ± 237.9 | 47.8 ± **32.9** | 187.4 ± 217.2 |
| button-press-topdown-v2 | **3411.8** ± 391.6 | 1567.5 ± 859.5 | 2224.3 ± 407.4 | 2084.9 ± **279.7** |
| coffee-button-v2 | **2112.7** ± 201.4 | 593.1 ± 453.4 | 739.6 ± 334.9 | 1374.3 ± 244.9 |
| pick-out-of-hole-v2 | **2759.6** ± 155.3 | 2743.7 ± 171.9 | 2235.0 ± 298.7 | 2187.1 ± **139.2** |
| assembly-v2 | **2837.8** ± 421.1 | 1313.9 ± 568.7 | 1277.1 ± **216.3** | 1798.9 ± 342.0 |
| reach-wall-v2 | **3488.0** ± **597.6** | 2427.3 ± 1078.1 | 2096.6 ± 882.6 | 2029.2 ± 1022.7 |
| plate-slide-v2 | **3945.4** ± 374.3 | 3112.0 ± **200.2** | 2649.0 ± 328.1 | 2806.8 ± 1080.0 |
| window-close-v2 | **1785.5** ± 369.1 | 177.8 ± **77.1** | 434.0 ± 309.4 | 319.1 ± 261.2 |
| handle-press-side-v2 | **1525.8** ± **47.1** | 1056.8 ± 193.8 | 1085.6 ± 67.0 | 1066.1 ± 148.6 |
| shelf-place-v2 | 5.1 ± **3.7** | 9.9 ± 5.3 | 13.4 ± 3.7 | **18.2** ± 6.6 |
| stick-pull-v2 | 6.7 ± **1.9** | 11.7 ± 6.5 | 14.1 ± 4.5 | **19.7** ± 8.0 |
| lever-pull-v2 | **3413.5** ± 269.0 | 898.5 ± 380.0 | 1088.5 ± **268.1** | 1873.7 ± 526.2 |
| button-press-v2 | **2928.9** ± **115.8** | 1133.1 ± 200.5 | 1315.3 ± 199.3 | 1483.0 ± 403.9 |
| pick-place-v2 | **740.1** ± 93.3 | 286.4 ± 250.8 | 133.8 ± **58.5** | 242.8 ± 62.8 |
| drawer-close-v2 | **3927.4** ± **276.2** | 1329.7 ± 1060.6 | 1592.4 ± 777.4 | 2335.4 ± 376.7 |
| door-open-v2 | **2364.4** ± 890.6 | 455.6 ± 318.6 | 488.5 ± **206.8** | 647.8 ± 326.5 |
| coffee-pull-v2 | 3580.8 ± 538.3 | 2529.4 ± 1162.6 | 3342.5 ± **214.7** | **3728.0** ± 261.4 |
| reach-v2 | **4406.6** ± **28.4** | 2325.8 ± 1121.4 | 3132.0 ± 1030.5 | 3890.6 ± 357.0 |
| hand-insert-v2 | **123.2** ± 28.6 | 74.7 ± 58.2 | 52.6 ± 21.9 | 53.5 ± **19.0** |
| door-unlock-v2 | **190.9** ± 35.7 | 156.7 ± 16.5 | 133.7 ± **12.8** | 140.9 ± 17.3 |
| box-close-v2 | 414.2 ± 145.5 | 184.3 ± 105.8 | **482.9** ± 197.2 | 256.8 ± **72.9** |
| bin-picking-v2 | **2038.7** ± **159.4** | 1432.4 ± 421.4 | 1647.6 ± 262.4 | 1411.4 ± 313.3 |
| door-lock-v2 | 1856.7 ± 318.1 | 1802.7 ± 473.8 | **2011.3** ± 240.8 | 1684.3 ± **201.2** |

