# OpenReview forum: "Reflect-then-Correct: Rebalancing Task Optimization for Generalizable Meta-Reinforcement Learning via Distributional Value Error Reduction"
_ICML.cc/2026/Conference — ICML 2026 regular_

### Official Review · Reviewer_SR12 · 2026-02-19

**Soundness:** 3
**Presentation:** 2
**Significance:** 3
**Originality:** 3
**Overall Recommendation:** 5
**Confidence:** 3

**Summary:**

The authors propose Relabel-then-Correct (RTC) that places a distributional view on task-based Meta-RL methods, and apply Sinkhorn TD for updating the meta-value distribution together with a recursive error correction module, to achieve improved performance in Metaworld ML10 and ML45. Ablation studies are performed to show both Sinkhorn and error correction are essential.

**Compliance With Llm Reviewing Policy:**

Affirmed.

**Final Justification:**

The rebuttal has addressed most of the concerns. I've raised the score from 4 to 5.

**Key Questions For Authors:**

- Would it be possible to have clearer definitions of mathematical terms?
- Roughly how long does it take to perform these Sinkhorn updates?
- What are the details of the meta-rl task representation learner? It seems to me this is only mentioned in line 15 of Algorithm 1, which is a PEARL / VariBAD like ELBO of trajectories. I think it's necessary to an extra section to elaborate on this and mention these methods.

I think RTC is an interesting contribution to the Meta-RL community, and would raise the mark if my concerns are addressed.

**Limitations:**

It would be interesting to see how RTC integrates with ELBO-free methods e.g., https://arxiv.org/abs/2512.20974

I'm not sure how the Sinkhorn algorithm is implemented, but as it is not mentioned in the paper I would like to point out https://arxiv.org/abs/2202.12404 that might lead to further speed up / memory efficiency.

**Strengths And Weaknesses:**

**strengths**:
- The motivation of considering distributional meta-values is clear and well justified by the experiments. Although the distributional RL / Sinkhorn part is not new, the error correction module is novel. The integration between distributional RL and Meta-RL is also interesting.
- The ablation study provides empirical evidence on the effectiveness of both components in RTC.

**weaknesses**:
- The main weakness is the presentation of the paper.rigorous and clear definitions of many terms are missing. A non-exaustive list of examples are: there is no mentioning of standard bellman backup, distributional bellman backup in non-MetaRL settings (they are familiar to everyone, but please have them in the paper for completeness); the reward function with task $r(s, a, z)$ is not defined at all; Eq. 4 is replaced with the mini-batch sampling form, what exactly is that form? The scalar forms $c_{i,j}, K_{i,j}$ are introduced but never appeared in the matrix-formed equations.

---

> ### Author Rebuttal · Authors · 2026-03-31
>
> We thank the reviewer for recognizing our work. We respond to each point below, and the results of integrating RTC with the ELBO-free method are shown in Fig.9 at: https://anonymous.4open.science/r/RTC2603-F724/
>
> **W1\&Q1:**
>
> Thanks for your advice. We provide clearer definitions of the mathematical terms below and will add them in the revision.
>
> 1. Standard and distributional Bellman backups in non-MetaRL settings:
> In standard RL, the action-value function is defined as $Q^\pi(s, a):=\mathbb{E}[Z^\pi(s, a)]=\mathbb{E}[\sum_{t=0}^{\infty}\gamma^t r(s_t, a_t)],s_{t+1}\sim p(\cdot\mid s_t,a_t)$, which estimates the expected value of $Z^\pi$. The standard Bellman operator [1] $\mathcal{F}^\pi$ is defined as $\mathcal{F}^\pi Q(s,a):=\mathbb{E}[r(s,a)]+\gamma\mathbb{E}_{p,\pi}[Q(s',a')]$. In contrast to expectation-based value estimation, distributional RL models the full return distribution. The corresponding distributional Bellman operator [Bellemare et al., ICML'17] $\mathcal{T}^\pi$ is defined as $\mathcal{T}^\pi Z(s,a):=r(s,a)+\gamma p^\pi Z(s,a)$, where $Z(s,a)$ denotes the value distribution and $p^\pi$ denotes the transition operator.
>
> [1] Bellman, R. (1957). Dynamic Programming.
>
> 2. In Eq.(3), $r(s,a,z)$ is a task-aware reward function.
>
> 3. Following [Genevay et al., AISTATS'18], Eq.(5) is the exact mini-batch sampling form, where sequences of meta-values and their TD targets, each of length $N$, are sampled to compute the cost function (Lines 206-216).
>
> 4. The scalars $c_{i,j}$ and $\mathcal{K}_{i,j}$ denote the $(i,j)$-th elements of the matrices $c$ in Eq.(5) and $\mathcal{K}$ in Eq.(6), respectively.
>
> **Q3:**
>
> Yes, as formulated in Line 15 of Algorithm 1, our meta-RL task representation learner follows a PEARL/variBAD-like ELBO over trajectories. The term $\mathbb{E}_ {g_\eta}[\log p(\tau_{:H}|\mathbf{z})]$ corresponds to the reconstruction loss, which is implemented by minimizing the prediction loss of the state decoder $d_s$ and the reward decoder $d_r$ [MetaDiffuser, ICML'23], namely $-\mathbb{E}_ {(s,a,s',r)\sim\tau_i}[\mathbb{E}_ {g_ \eta}[\log d_s(\hat{s}'|s,a,\mathbf{z})+\log d_r(\hat{r}|s,a,\mathbf{z})]]$. The KL term denotes the divergence between the variational posterior $g_\eta$ and the Gaussian prior $p(\mathbf{z})$. Together, these terms encourage the learned task representations $\mathbf{z}$ to focus on task-relevant information and to distinguish task changes under a unified learning objective.
>
> **L1:**
>
> Following your suggestion, we conducted new experiments integrating RTC with the recommended ELBO-free method GLiBRL to further validate our approach. We experimented on the more challenging ML-10 setting, where the agent is trained only on 10 tasks and must adapt to 5 unseen test tasks. As illustrated in Fig.9 (see link), GLiBRL+RTC achieves superior test performance compared to GLiBRL. This indicates that the Sinkhorn distributional meta-value learning and error correction mechanism potentially lead to more accurate meta-value updates, thereby improving the adaptability of the meta-policy.
>
> **Q2\&L2:**
>
> Thanks for your advice. Our Sinkhorn updates add only a modest amount of training time: training takes 177.6 hours with the Sinkhorn update on ML-10, compared to 163.2 hours without it. For our future work, we will consider the paper provided by the reviewer to further speed up the Sinkhorn iteration.
>
> We implement the Sinkhorn algorithm as follows. As in Eq.(5), we first compute the cost matrix $c$ between two sets of meta-values and their targets using the squared Euclidean distance. We then initialize two uniform marginal distributions along with their corresponding scaling vectors $u$ and $b$. At each iteration, the scaling vectors are updated according to Eq.(7), allowing the transport matrix $P$ to progressively satisfy the marginal constraints on both sides. The iteration terminates when the changes in the scaling vectors fall below a predefined threshold. The approximate optimal transport matrix $\hat{P}^\star$ is then constructed from the final scaling vectors according to Eq.(6). Substituting $\hat{P}^\star$ and $c$ into Eq.(5), we obtain the Sinkhorn TD error $\mathcal{C}$. We have also released the source code (Line 708) to facilitate implementation.

---

> > ### Author Rebuttal · Reviewer_SR12 · 2026-04-01
> >
> > I appreciate the additional experiments, and I assume the quality of the presentation can be improved from addressing comments from reviewer 2sQR. I've raised the score to 5.
> >
> > However, I do see a mismatch for the reported GLiBRL results. I see a success rate drop by 10% in the reported result (compared to the reported ~25% in ML10 using 2e7 steps). Can the author clarify if this is from the use of value functions, as the original GLiBRL does not learn value functions at all, or is it because of something else?

---

> > > ### Author Response · Authors · 2026-04-01
> > >
> > > Thank you for your feedback and for raising the score to 5.
> > >
> > > Yes, the original GLiBRL does not learn value functions. Inspired by AMAGO-2, which leverages distributional RL to convert real-valued meta-value targets with varying return scales into classification labels, we introduce a learnable value network to model the meta-value function in GLiBRL using our Sinkhorn TD update to deal with diverse meta-tasks.
> > >
> > > The mismatch in the reported GLiBRL results is likely due to experimental variance. Due to time and computational resource constraints, we evaluated only three random seeds. We did not initially notice that the GLiBRL paper reports results averaged over ten seeds, which may lead to more stable and slightly higher performance. We will further verify our revised code and run experiments with more seeds. We plan to provide additional evaluations and update the new results at the provided link before the end of the discussion period.
> > >
> > > -- promised another round of reply/experiments before ddl --
> > >
> > > Since our last discussion, we have extended our experiments with more seeds and longer training horizons. Unfortunately, We observed that the integration of GLiBRL+RTC exhibits significant instability, with average success rates occasionally collapsing. It is worth noting that this high variance is somewhat consistent with the original GLiBRL paper (as seen in Fig.1).
> > >
> > > We hypothesize this stems from a representation mismatch: GLiBRL utilizes a 392-dimensional belief distribution as a task representation, which vastly outweighs the 39-dimensional state information. We conjecture that during value network training, the state’s signal is numerically overwhelmed by the high-dimensional task representation, leading to degenerate value estimates. While the prior framework was elegant in its avoidance of such networks, our initial assumption that a value network could be seamlessly integrated now appears overly premature. A forced introduction of a value network may indeed compromise the structural integrity of GLiBRL. We appreciate the reviewer for bringing this work to our attention.
> > >
> > > Furthermore, to ensure a rigorous and fair comparison, we evaluated GLiBRL using the identical seed configurations employed for the main results in Fig.2 of our paper, and the newly updated results are detailed in Fig.11 at the provided link. The results confirm that our RTC method consistently outperforms GLiBRL, demonstrating the efficacy of our distributional meta-value learning.

---

### Official Review · Reviewer_2sQR · 2026-03-09

**Soundness:** 3
**Presentation:** 2
**Significance:** 2
**Originality:** 3
**Overall Recommendation:** 3
**Confidence:** 4

**Summary:**

This work proposes RTC, a meta-reinforcement learning algorithm which trains a distributional value function using a Sinkhorn divergence-based objective (Sinkhorn TD) together with an error-aware reweighting mechanism intended to stabilise training. RTC is evaluated on the ML10 and ML45 settings of the MetaWorld benchmark, where it is shown to outperform considered baselines and the contribution of both its components is examined through ablations.

**Compliance With Llm Reviewing Policy:**

Affirmed.

**Final Justification:**

The work presents RTC, a metaRL method designed to deal with disparate return distributions in settings with non-parametric task variability that induce them. The method's mechanism for doing so--using a Sinkhorn TD objective with a distributional value network--is intuitively sound and specifically within the metaRL setting it is novel. The addition of the error correction mechanism appears theoretically justified and novel. RTC seems to modestly outperform baselines in Meta-World's challenging ML10 and ML45 settings, with reasonable experimental methodology.

The work's primary weaknesses lie in its presentation. The clarity of the text leaves much to be desired, particularly in the rigour of the mathematical presentation of the method and the novel error correction mechanism. More importantly, the significance of the work for the broader metaRL field can be contested as only a single benchmark is used and the method proposed targets a hyper specific detail of this benchmark--the disparate task returns. This is an implementation detail in Meta-World and is not necessarily indicative of a problem that's theoretically crucial to non-parametric task variability; tasks can meaningfully vary while their returns also lie in a similar range, and in fact one could simply normalise the returns in Meta-World to achieve a similar effect. Nevertheless the proposed Sinkhorn-divergence-based objective is almost entirely focused on mitigating this shortcoming, and thus the work over-indexes on what is not necessarily a truly important problem in meta-reinforcement learning. The work lacks additional justification and motivation for why resolving this problem is impactful.

The authors provided a thorough rebuttal to my concerns, primarily addressing requests for clarity regrading details of their experiment methodology, and notably provided adjusted statistical significance tests that were requested. While the additional clarification in the rebuttal and additional results were appreciated, the core shortcoming of the work was not sufficiently addressed, and in addition it is unclear whether the authors would perform the much needed significant improvement of the paper's presentation for the camera-ready version, especially as they did not seem to acknowledge the higher level issues with their mathematical rigour following Equation 8. As a result, my score remains unchanged after the rebuttal, which reinforced my prior assessment and I cannot recommend the paper for acceptance as it stands.

**Key Questions For Authors:**

1. How exactly is the meta-value $\tilde{v}_\phi$ parameterised and sampled from in practice?
2. What exact RL algorithm is used for the policy update in Equation 1?
3. Can the authors clarify the intended behaviour of the error-correction weighting scheme in Section 4.2? Is it to weigh down high-error samples as presented?

Addressing these points would greatly improve the presentation of the paper.

4. Can the authors provide stronger evidence that the benefits of RTC are not specific to MetaWorld's variance of return magnitudes? For example, results in other meta-RL benchmarks. This would help strengthen the significance of RTC's algorithmic contributions.
5. Can the authors provide evidence to support their claims for the favourable optimisation or generalisation dynamics of Sinkhorn TD? For instance, additional citations, theoretical results or qualitative analyses.

**Limitations:**

yes

**Strengths And Weaknesses:**

Strengths
- The application of Sinkhorn divergence to training critic networks in MetaRL appears novel and is relevant when return distributions are highly mismatched across tasks. The error correction component of RTC appears novel and is supported by a nontrivial theoretical motivation. The overall method is interesting and the high-level technical formulation appears sound.
- RTC appears to outperform baselines in MetaWorld's challenging non-parametric meta-learning setting.
- The provided ablation supports that both architectural contributions of the method are important for its performance.
- The authors provide source code, which strongly supports the reproducibility of the method and helps answer questions about undisclosed implementation details.

Weaknesses
- The work only uses MetaWorld as a benchmark, which limits the method's potential significance to the wider MetaRL landscape.
  - A central motivation for the method is the ability to deal with widely varying return distributions, which is a particularly salient in MetaWorld and is present even in its multi-task (and not meta-learning) settings. It remains unclear if this is also an issue present in other MetaRL benchmarks, and therefore whether the proposed method yields similar performance gains in other settings.
- The experimental methodology could use additional rigour, without which it is hard to gauge whether the modest performance increase in MetaWorld is significant.
  - The work only uses four seeds which might be insufficient for MetaWorld's high-variance MetaRL setting, and reports mean and standard deviation, metrics which are not resistant to outlier seeds. More seeds (e.g. 5-10 as tends to be standard for the benchmark) with inter-quartile mean and inter-quartile range (as methodologically proposed by [1]) reporting would strengthen the statistical significance of the results.
  - The version of MetaWorld used is not clear from the text or the provided code. This can be an issue for comparing to past works in the literature, or other works comparing to this work in the future, as outlined by [2]. A concrete benchmark git commit hash in the Appendix would be helpful.
  - It is unclear which implementations of the baselines the authors used in their work, or whether the results have been taken from the baselines' original publications. From the provided code, it is unclear which baselines were re-run in this work.
  - The provided implementation seems to suggest that the authors are using a custom evaluation routine for meta-learning on MetaWorld, but this routine is not described in detail in the experiment setup section or ideally an Appendix. Crucially: how many trials are used per evaluation test task? Does each trial have a different goal position? These details are not provided.
  - The hyper parameters listed in Table 3 are assumedly for RTC. Presumably hyper parameters present here, notably learning rate and entropy term coefficient, were not tuned for each method. Since each baseline has a different architecture or at least loss landscape resulting from its different objective, certain hyper parameters should ideally be tuned per method to strengthen the baselines and therefore also strengthen RTC's results against them, or the paper should justify why shared settings are sufficient.
- The results of the qualitative analysis of the learned task representations are unconvincing.
  -  The authors claim that RTC exhibits clearly separated clusters, but so do the clusters of the rest of the baselines. In Figure 3, there appears to rather little overlap between lever-pull and sweep-into, or lever-pull and drawer-open, but it is unclear that this overlap is enough to characterise the representations learned by the baselines as insufficient.
  - Additionally, in Figure 4, while the heat map of task representations does support the claim that similar tasks receive similar representations, it's unclear how RTC's contributions in particular contribute to this. A comparison with similar heatmaps from other baselines should be provided in the Appendix.
- The paper's presentation can often lack clarity.
  - The parameterisation of the distributional meta-value network, and notably how particles are sampled from it, is not discussed in the main text or any Appendix.
  - The paper does not provide details on the network sizes used for RTC and the corresponding baselines. Appendix B.3 mentions that the architectures and layer widths are kept consistent across methods, but that is not possible, since e.g. AMAGO2 is a Transformer, while VariBAD is using a sequence-model (GRU) VAE.
  - Algorithm 1 and Equation 13 do not make it clear whether the target branch is detached, or whether it uses a target network, or how gradients are propagated through the Sinkhorn block. The use of a target network or the stop-gradient operator in the target branch should be explicitly specified.
  - In Algorithm 1 Line 14, it's mentioned that the policy is updated with Eq. 1, but Eq 1 is not a tractable optimisation objective.
  - The abstract mentions that the adaptive importance weights prioritise transitions critical for accuracy. Section 4.2 also mentions that "This error-aware reweighing enables the target correction of hard tasks". These two statements imply that transitions that lead to low accuracy (high error) should be prioritised. The section is further written to read as though the idea is to prioritise high-error samples, until Theorem 4.2 concludes otherwise: Equation 11 defines that value updates are weighted inversely-proportional to their Sinkhorn TD error, and Equation 12 does so as well. Appendix A further states that the weighting "induces a conservative distribution that prioritises reliable samples". This appears in contradiction to prose earlier in the text. Readjusting phrasing across the text would help prevent this tension.
  - Figure 6 attempts to visualise the loss landscape of the model parameters, but it is unclear how this plot has been generated. What are alpha and beta?
- The presentation also often lacks notational precision.
  - Equation 3, albeit intuitive, is written too informally, while it's not deliberately stated to be informal. Concretely the randomness over $a$ and $s^\prime$ should be specified, as well as whether this is equality of random variables or equality in distribution.
  - In Equations 5 and 7, $\mathbf{1}_N$ (presumably taken to be the all-ones vector with N elements) is not defined, and more importantly the normalisation convention for the empirical measures is not discussed. This makes it unclear how the sampled particles are weighted.
  - Equation 9 uses $\mathcal{P}^{\pi_{k-1}}\delta_{k-1}$ but Equations 10 and 12 use $P^{\pi_{k-1}}\delta_{k-1}$, which is likely merely notational drift.
  - In the definition of $\mathcal{P}^{\pi}\delta(s,\mathbf{z}):=\mathbb{E}_{s^\prime\sim(\cdot|s,a),a^\prime\sim\pi(a|s,\mathbf{z})}$, $a^\prime$ is likely meant to be $a\sim\pi(\cdot|s,\mathbf{z})$.
  - Equation 8 is key for the intuition of a core contribution of the method, but is currently overly terse.
  - Equation 12 defines the practical weights only up to proportionality, but Equation 13 takes $w_k(s,\mathbf{z})$ as though it is a fully specified quantity. The paper does not describe how these weights are instantiated in practice before used in the objective.
  - In Equation 14, $\delta_{k-1}$ appears in the target, but it is unclear whether this denotes a  detached previous iterate e.g. $\delta_{\psi_{old}}$, a target error network of sorts, or another stored estimate.
- Certain claims in the text are not properly substantiated
  - Section 4.2 claims that the error correction "enables the targeted correction of hard tasks and improves generalisation in the meta-RL setting", but there is no result provided that directly supports this claim. The ablation that, without the error correction, the performance of RTC either incurs higher variance or modestly drops in Figure 5, which perhaps indirectly supports this, but is insufficient to directly validate the improved generalisation claim. The mechanism through which it would improve generalisation (targeted correction of hard tasks) is also not what the method appears to implement as discussed previously.
  - The final paragraph of Section 4.1 claims that a key advantage of using Sinkhorn TD is its ability to "align meta-value distributions across different tasks" but this is not substantiated with results proving that training under this objective empirically or theoretically induces a shared geometry in parameter space. The clearest supporting advantage of the method seems to be the use of a flexible distributional critic without fixed support, which enables return distributions of different magnitudes across tasks to be modelled. By contrast, the claimed cross-task alignment effect of the Sinkhorn TD objective is not established by the objective itself or empirical results.
  - The same paragraph also suggests that cross-task gradient interference is also mitigated but this is additionally not supported by any empirical results.

Minor comments
- The geometric language throughout the text is evocative but not clearly connected to the concrete algorithmic choices or quantities in the implementation. Specifically, phrases in the introduction such as "induce a Riemannian geometry that morphs disparate task returns into a unified metric space via an interpolative gradient flow" come to mind. While the underlying intuition is plausible, such language should be limited, as it mostly serves to obfuscate the approach.
- The work uses the term "meta-value" liberally throughout the text but it is never defined. It can be taken to mean "the value function, when training in the meta-RL setting" but in that case simply using the term "value" would be more readable. The use of the "meta-" prefix is not standard in the literature. Similar for the use of the term "meta-policy".

References
- [1] Agarwal, Rishabh, et al. "Deep reinforcement learning at the edge of the statistical precipice." Advances in neural information processing systems 34 (2021): 29304-29320.
- [2] McLean, Reginald, et al. "Meta-world+: An improved, standardized, rl benchmark." arXiv preprint arXiv:2505.11289 (2025).

---

> ### Author Rebuttal · Authors · 2026-03-31
>
> New interquartile mean (IQM) (Fig.8), hard task (Tab.9) and heatmap results (Fig.10) are provided here: https://anonymous.4open.science/r/RTC2603-F724/
>
> # W1\&Q4
>
> We clarify that our paper focuses on more challenging non-parametric settings with fundamentally different tasks. Currently, ML-10 and ML-45 in Meta-World are widely accepted as the most standard benchmarks for evaluation under such setups [SDVT; Beck et al., 2023].
>
> # W2.1
>
> For a fair comparison, we used the same random seeds across all methods. The new IQM results demonstrate that our results are statistically significant.
>
> # W2.2
>
> We provided the version of Meta-World [Yu et al., 2019] in Line 57, and all the methods were compared on the same version.
>
> # W2.3
>
> **We clarify that all results are NOT taken from the original publications; we re-ran all baselines and provided the training curves in Fig.2.**
>
> # W2.4
>
> Following SDVT, we used 50 trials per test task, with each trial having a different goal position.
>
> # W2.5
>
> Tab.3 presents only the hyperparameters of our RTC. For the baselines, we tuned and adopted the appropriate hyperparameters based on their source code.
>
> # W3.1
>
> In Fig.3, all the baselines exhibit overlapping clusters or mixed colors within clusters, whereas only our RTC forms fully separated, single-colored clusters, indicating that the baselines cannot accurately distinguish between different tasks.
>
> # W3.2
>
> The Sinkhorn TD update and error correction in RTC yield more accurate meta-value estimates, enabling the collection of more task-specific trajectories to infer more discriminative task representations. Heatmaps of baselines in new experiments further show that RTC learns better shared representations.
>
> # W4.1\&Q1
>
> Meta-value $\tilde{v}_\phi$ is parameterized by an MLP that outputs samples (referred to as particles) to approximate the value distribution and is updated using the Sinkhorn TD error (Eq.5).
>
> # W4.2
>
> Network size details are in Tab.4. Following the SDVT source code, the architecture and layer widths, except for AMAGO-2, are kept consistent across RTC and other baselines. AMAGO-2 was implemented according to its source code using a transformer. We'll revise.
>
> # W4.3
>
> We adopted PPO as the backbone for updating the policy and value networks, which do not use target networks.
>
> # W4.4\&Q2
>
> We have indicated in Tab.3 that our policy algorithm is PPO.
>
> # W4.5\&Q3
>
> The error-correction weighting scheme is trained to reduce the weight of high-error samples. Our claim is also consistent with the correction of hard tasks, as hard tasks generally have higher estimation errors, especially during early training. We'll revise the phrasing accordingly.
>
> # W4.6
>
> We adopted [1] to visualize the landscape of $\tilde{v}_\phi$. Let $d_1, d_2$ be two directions in parameter space, and let alpha and beta be scalars controlling the step size. The perturbed parameters are defined as $\phi(alpha,beta)=\phi+alpha\cdot d_1+beta\cdot d_2$. At each grid point, we compute the loss $L(\phi(alpha,beta))$.
>
> # W5.1
>
> We'll add $a\sim\pi(\cdot|s,\mathbf{z})$ and $s'\sim p(\cdot|s,a)$ to Eq.(3).
>
> # W5.2
>
> $\mathbf{1}_N$ is a vector of ones. Eq.(7) performs iterative normalization, ensuring that the transport matrix satisfies the constraint that both its row and column sums equal one.
>
> # W5.3
>
> Typo. Should be $\mathcal{P}$ in Eq.(10) and (12).
>
> # W5.4
>
> Typo. Should be $a$ in Line 256.
>
> # W5.5
>
> We have not found any typos in Eq.(8).
>
> # W5.6
>
> In practice, before applying Eq.(13), we normalize the approximate proportional weights from Eq.(12) using a softmax function to preserve their relative importance.
>
> # W5.7
>
> Similar to DQN, $\delta_{k-1}$ is implemented as a target network to stabilize training and is updated separately using an EMA of $\psi$.
>
> # W6.1
>
> The improved generalization is supported by Fig.6. With error correction, the landscape becomes smoother and flatter, which is often related to better generalization [1]. Moreover, Tab.9 shows that RTC with error correction achieves significantly higher returns on many hard tasks.
>
> # W6.2\&Q5
>
> We clarify that the alignment of meta-value distributions is an intrinsic property of the Sinkhorn TD objective, rooted in the foundations of Optimal Transport (OT) [Genevay et al., 2018]. AMAGO-2 shows that modeling value distributions helps handle meta-tasks with different return scales. Sinkhorn divergence provides a principled and efficient approximation of OT. Unlike the fixed supports used in AMAGO-2, Sinkhorn divergence compares distributions directly within the underlying geometry, naturally accommodating value distributions with different magnitudes across tasks.
>
> # W6.3
>
> As in Tabs.6,7,and 9, RTC achieves the highest return on most tasks, particularly on harder tasks. This suggests that updates across tasks may be more aligned, helping to mitigate cross-task gradient interference.
>
> [1] Visualizing the loss landscape of neural nets. NeurIPS'18.
>
> In Line 163, we defined the meta-value $\tilde{v}$.

---

> > ### Author Rebuttal · Reviewer_2sQR · 2026-04-03
> >
> > The detailed response is appreciated. In particular the IQM results are good to have.
> >
> > > We clarify that our paper focuses on more challenging non-parametric settings with fundamentally different tasks. Currently, ML-10 and ML-45 in Meta-World are widely accepted as the most standard benchmarks for evaluation under such setups [SDVT; Beck et al., 2023].
> >
> > While the use of Meta-World is understandable because it is true it is a primary non-parametric task distribution meta-rl benchmark, my point still stands. The work claims to aid in non-parametric task distribution settings in general, while the approach proposed is essentially targeting a technicality that is very specific to Meta-World (namely, large disparity in returns from different tasks).
> >
> > > We clarify that all results are NOT taken from the original publications; we re-ran all baselines and provided the training curves in Fig.2.
> >
> > Could you clarify which exact implementations were used for each method in that case, given that the provided code does not , if I'm not mistaken, provide implementations for all baselines present?
> >
> > > Tab.3 presents only the hyperparameters of our RTC. For the baselines, we tuned and adopted the appropriate hyperparameters based on their source code.
> >
> > This detail should be provided in the Appendix.
> >
> > > We adopted PPO as the backbone for updating the policy and value networks, which do not use target networks.
> >
> > > Similar to DQN, $\delta_{k-1}$ is implemented as a target network to stabilize training
> >
> > These statements are contradictory. From what I surmise, the latter statement is to be taken as true. This detail should also be added to the main text for clarity, precisely *because* PPO normally doesn't have target networks.
> >
> > > We have indicated in Tab.3 that our policy algorithm is PPO.
> >
> > Thank you for pointing that out. Such a detail should be present in the main text to strengthen the presentation. The general RL objective present is currently too opaque.
> >
> > > We adopted [1] to visualize the landscape
> >
> > Thank you for clarifying. This should be detailed in the figure's label.
> >
> > > We have not found any typos in Eq.(8).
> >
> > To clarify, I said that it is terse, not that it has any typos. This is of course subjective but I believe it is currently difficult for the reader to parse, which is a weakness when that equation is key to understanding RTC. Other reviewers have made similar comments regarding the paper's presentation in this section, with reviewer y6Un in particular mentioning this exact equation as well as what follows. There are many possible ways to address this issue, such as providing additional commentary on it, although that might be difficult due to space constraints.
> >
> > Overall, the effort to address my concerns is much appreciated but for the time being I maintain my score. The paper's presentation leaves a lot to be desired and the text needs relatively major revisions, while the primary contributions of RTC to the field more broadly still remain unconvincing.

---

> > > ### Author Response · Authors · 2026-04-08
> > >
> > > # Q3 & Q5 & Q6
> > >
> > > Thank you for your concrete suggestions. We will update the paper accordingly, including moving the description of the PPO algorithm from the Appendix to the main text, providing baseline hyperparameters in the Appendix, and clarifying the labels in Fig.6.
> > >
> > > # Q1
> > >
> > > We respectfully clarify that our proposed approach is not specifically designed for Meta-World. Recent research on non-parametric meta-tasks has received little attention due to the higher demands on algorithmic adaptability and computational resources. Consistent with prior art such as SDVT, we utilize the ML-10 and ML-45 benchmarks from Meta-World for evaluation. Beyond the inherent challenge of large return disparities, these benchmarks are characterized by significant semantic heterogeneity. For instance, tasks range from door-closing to plate-retrieving, involving distinct contact dynamics and functional objectives. In our setup, the agent is trained on 10 (ML-10) or 45 (ML-45) meta-training tasks and subsequently evaluated on 5 unseen test tasks. This broad semantic span provides a rigorous testbed for assessing an agent's generalizability and adaptability across functionally disparate domains.
> > >
> > > # Q2
> > >
> > > We provided the source code link of our proposed method RTC in Line 708. We implemented SDVT, LDM, and variBAD based on the source code released by the SDVT paper. For AMAGO-2, we used the official source code provided by the AMAGO-2 paper.
> > >
> > > # Q4
> > >
> > > We respectively clarify that these statements are NOT contradictory. We employ PPO as the backbone, and the value and policy networks do not use target networks. The additionally introduced error model, $\delta_\psi$, is used to weight samples when updating the value network. We only include a target network for the error model to ensure stable updates.
> > >
> > > # Q7
> > >
> > > Eq.(8) is the optimization objective for learning the meta-value $\tilde{v}_k$, which aims to minimize the distance between the learned meta-value $\tilde{v}_k$ and the true optimal value $\tilde{v}^\star$ (first line of Eq. (8)). However, in RL, the true optimal $\tilde{v}^\star$ is not accessible. Value estimation is therefore generally used to approximate the optimal value [Sutton \& Barto, 1998]. The second line of Eq.(8) is to choose the estimated value with the minimal Bellman estimation error. And the sum of training distribution $p_k$ should equals to one.
> > >
> > > The Rebuttal Acknowledgement by Reviewer y6Un has shown that we have fully resolved the concerns, including Eq.(8).

---

### Official Review · Reviewer_JfTG · 2026-03-10

**Soundness:** 3
**Presentation:** 3
**Significance:** 3
**Originality:** 3
**Overall Recommendation:** 4
**Confidence:** 4

**Summary:**

This paper addresses the challenge in meta-reinforcement learning where heterogeneous tasks exhibit different reward scales, leading to imbalanced optimization and poor generalization. The authors propose RTC (Reflect-then-Correct), a two-stage framework that first aligns value distributions across tasks using Sinkhorn distance to decouple reward scale discrepancies, and then corrects value estimation bias via a recursive Bellman error model that adaptively reweights critical transitions. Experiments on challenging Meta-World benchmarks (ML-10/ML-45) show that RTC achieves state-of-the-art performance in both training and test success rates.

**Compliance With Llm Reviewing Policy:**

Affirmed.

**Final Justification:**

Thanks for your rebuttal. All my concerns are resolved, and I keep my positive score.

**Key Questions For Authors:**

See Weakness above.

**Limitations:**

See Weakness above.

**Strengths And Weaknesses:**

**Strengths:**

- The use of Sinkhorn distance for geometric alignment of value distributions effectively mitigates reward-scale imbalance, while the error-correction module identifies high-impact transitions and applies importance weighting to reduce sampling bias. The integration of these two components is conceptually clean and well-motivated.
- Empirical results are strong: RTC outperforms recent baselines on standard meta-RL benchmarks, and learned task representations demonstrate improved cluster separation, suggesting better task disentanglement.

**Weaknesses:**

- The computational overhead of Sinkhorn iterations and the error model is non-negligible, yet the paper does not report training time, memory usage, or sample efficiency compared to baselines, making it difficult to assess practical efficiency.
- There is no ablation study or sensitivity analysis on the key hyperparameter $\lambda$ (entropy regularization coefficient) in the Sinkhorn algorithm, which critically affects the trade-off between accuracy and smoothness.
- The related work section lacks discussion of recent advances in distributional RL and optimal transport–based value alignment in meta-RL (e.g., works leveraging Wasserstein/Sinkhorn distances for policy or value matching). This omission weakens the positioning of the method’s novelty and makes it hard to evaluate the true technical contribution.

---

> ### Author Rebuttal · Authors · 2026-03-31
>
> **W1: Comparison of computational overhead**
>
> **The Sinkhorn iterations and the error model of RTC add only a modest computational cost.** Using the ML-10 as examples, RTC stands out with the lower memory usage (Mb) and a higher Frames Per Second (FPS) (number of steps/s) compared to most competitive meta-RL methods. Despite incorporating the Sinkhorn iterations and error model, our RTC’s computational overhead does not substantially surpass that of variBAD.
>
> **Table 10: Computational overhead comparison among different methods.**
>
> |Metric|RTC|SDVT|LDM|variBAD|AMAGO-2|
> |-|-|-|-|-|-|
> |Mb|1640|1376|**1352**|1738|6808|
> |steps/s|320|293|314|**337**|267|
>
> **W2: RTC is not sensitive to the key hyperparameter $\lambda$.**
>
> Following your suggestion, we conducted additional experiments to verify that RTC is relatively insensitive to the entropy regularization coefficient $\lambda$. As shown in Table 8, an appropriate $\lambda$ provides a trade-off between accuracy and smoothness and yields better convergence results. A larger $\lambda$ overly smooths the transport plan and introduces additional bias, whereas a smaller $\lambda$ leads to poorer stability and increases sensitivity to noise. Nevertheless, the final performance remains relatively insensitive to $\lambda$.
>
> **Table 8: Converged average test success rate ± standard error (\%) with different $\lambda$ values.**
>
> |Task|$\lambda=0.1$|$\lambda=1$ (RTC)|$\lambda=10$|
> |-|-|-|-|
> |ML-10|29.38±2.32|**30.39**±**1.62**|23.41±3.64|
>
> **W3: More discussions of related works**
>
> Recently, distributional RL and optimal transport–based value alignment in meta-RL have received relatively limited attention. We discussed AMAGO-2 in the Related Work section (Line 134), which is closely related to our RTC method and also investigates distributional value learning for meta-RL with fundamentally different tasks. However, discretization-based approaches rely on a predefined number of bins and value ranges, which may lead to inaccurate meta-value estimates when return scales vary significantly. To the best of our knowledge, RTC is the first to leverage Sinkhorn distances for meta-value learning, enabling adaptive modeling of meta-value distributions with different return scales in meta-RL settings. Furthermore, we introduce a recursive error model that prioritizes critical samples and mitigates the impact of statistical bias during Sinkhorn updates.
>
> Another line of meta-RL research integrates optimal transport techniques, such as the Wasserstein distance, for auxiliary purposes including measuring task distribution divergence [1], quantifying task similarity [2], designing intrinsic rewards [3], and selecting behavior policies for offline data collection [4]. Specifically, DiAMetR [1] uses the Wasserstein distance to measure divergence between meta-training and meta-testing task distributions. SimBelief [2] employs Wasserstein-based bisimulation metrics to compute distances between state-transition dynamics for task similarity estimation. WURL [3] proposes a Wasserstein distance–based intrinsic reward that maximizes the discrepancy between state distributions induced by different meta-policies to encourage exploration. ER-TRL [4] considers the offline meta-RL setting and computes the average Wasserstein distance between behavior policies to reduce their influence on learned task representations. Nevertheless, these methods do not address optimal transport–based policy or value matching in meta-RL.
>
> [1] Ajay A, et al. Distributionally adaptive meta reinforcement learning. NeurIPS 2022.
>
> [2] Zhang M, et al. Learning Task Belief Similarity with Latent Dynamics for Meta-Reinforcement Learning. ICLR 2025.
>
> [3] He S, et al. Wasserstein unsupervised reinforcement learning. AAAI 2022.
>
> [4] Nakhaeinezhadfard M, et al. Entropy regularized task representation learning for offline meta-reinforcement learning. AAAI 2025.
>
> Additional experimental results are provided here: https://anonymous.4open.science/r/RTC2603-F724/

---

> > ### Author Rebuttal · Reviewer_JfTG · 2026-04-02
> >
> > Thanks for your effort. I maintain my score.

---

> > > ### Author Response · Authors · 2026-04-03
> > >
> > > Dear Reviewer,
> > >
> > > Thank you very much for your feedback. We are pleased to learn that our rebuttal has fully resolved your concerns and that you will maintain your positive score. We will revise our paper accordingly to incorporate these points.
> > >
> > > Please let us know if you have any further questions; we would be happy to address them before the discussion period ends.
> > >
> > > Thank you again for your time.
> > >
> > > Best regards,
> > >
> > > The Authors

---

### Official Review · Reviewer_y6Un · 2026-03-13

**Soundness:** 2
**Presentation:** 4
**Significance:** 2
**Originality:** 2
**Overall Recommendation:** 3
**Confidence:** 3

**Summary:**

This paper studies generalizable meta-reinforcement learning in non-parametric task settings, where training and test tasks differ semantically and structurally, rather than only through small parameter perturbations. To address this, the paper proposes Reflect-then-Correct (RTC). The first component models the meta-value through Sinkhorn divergence, treating value distributions as adaptive floating particles instead of fixed-support categorical bins. The intended benefit is a geometry-aware alignment across tasks with heterogeneous return scales. The second component introduces a recursive error model to track temporal accumulation of Sinkhorn TD inconsistencies, and uses this to construct adaptive importance weights that reweight transitions toward samples deemed more critical for estimation accuracy. Empirically, the paper evaluates RTC on Meta-World ML-10 and ML-45, comparing against SDVT, LDM, variBAD, and AMAGO-2. The reported results show that RTC achieves the best average test success rate and test return on both benchmarks, with supporting ablations indicating that both the Sinkhorn distributional update and the error correction module matter.

**Compliance With Llm Reviewing Policy:**

Affirmed.

**Key Questions For Authors:**

1.How should the reader interpret the “theoretical guarantee” for the practical weighting rule in eq.12?

2. How accurate and stable is the learned error model \delta_{\psi} in practice?

3. What is the computational overhead of RTC relative to the baselines?

4. How robust are the gains across seeds and hyperparameters?

**Strengths And Weaknesses:**

pros:
1. the paper focuses on a relevant and nontrivial problem setting
2. the empirical results are promising
3. the ablations are good.

cons
1. the role of the error model is intuitive but not sharply justified.
The recursive error model \delta_{k} is introduced to capture temporal accumulation of Bellman inconsistencies, and that makes sense at a high level. But it is not entirely clear how accurately this auxiliary model can be learned in practice, or how sensitive the final weighting is to approximation error in \delta_{\psi}. Since the reweighting mechanism is central to the method, a more direct empirical analysis of the quality and stability of these weights is needed.

2. the comparison set is somewhat incomplete for the claim being made.
The baselines include variBAD, LDM, SDVT, and AMAGO-2, which are relevant. However, the paper’s strongest methodological contrast is really with distributional/meta-value learning under heterogeneous returns, and in that sense AMAGO-2 is the main direct comparison.

3. the theoretical section is not fully convincing as written.
The paper claims a theoretical guarantee for the reweighting strategy, but the path from the optimization problem in Eq. 8 to the practical weighting rule in Eq. 12 feels loose. In particular, the derivation of the optimal training distribution p_k uses a “relaxed version” of the original problem and then introduces a soft-min style reformulation in the appendix. The resulting exponential weighting rule is plausible as a heuristic, but I do not think the current presentation establishes a strong theorem in the usual sense. There are several approximations and substitutions along the way, including replacing the true discrepancy with the recursively estimated error model and collapsing constants into a temperature parameter \beta

---

> ### Author Rebuttal · Authors · 2026-03-31
>
> **W1Q2:**
>
> We appreciate this thoughtful concern. We address it from three perspectives: theoretical grounding, empirical validation, and additional experiments.
>
> (1) The bootstrapped target, rather than a ground-truth target, may cause the meta-value network to collect seemingly beneficial yet potentially misleading samples. According to Lemma 4.1 and Theorem 4.2, the fixed point of the Bellman-style error update in Eq.(14) approximates an upper bound of the true meta-value estimation error, ensuring that the error model reflects the underlying estimation bias. By treating the one-step estimation error as an immediate reward and recursively propagating it, the Bellman-style error training captures how future accumulated errors influence the current estimate. This helps identify high-error states that significantly affect subsequent value predictions and suppresses the gradual accumulation and amplification of errors. Furthermore, the target error model aims to reduce variance and stabilize learning, ultimately leading to more reliable weight sampling and mitigating the true value estimation error.
>
> (2) The ablation study in Fig.5 shows that error correction via sample reweighting further improves adaptability and final performance. This effect is particularly evident in the more challenging ML-10 setting, where the agent is trained on only 10 tasks and must adapt to 5 unseen test tasks; in this case, the error correction mechanism significantly reduces variance and enhances training stability. Moreover, the landscape of meta-value in Fig.6 demonstrates that the learned error model improves generalization and mitigates error accumulation.
>
> (3) We conducted further experiments to validate the quality and stability of the error model's output weights. When we delayed updates of the error model based on the recursive TD error and instead relied primarily on the TD error of the meta-value, the test success rate ± standard error (\%) results (**19.25±3.29**) showed a degradation in final performance compared to RTC (**29.87±3.18**). This indicates that the error model helps reduce estimation errors, whereas updating weights solely with bootstrapped target meta-values may fail to correct them, consistent with the first point.
>
> **W2:**
>
> We agree that AMAGO-2 is the most direct methodological comparison and appreciate the chance to sharpen this comparison.
>
> Distributional RL [Bellemare et al., ICML’17] preserves multimodality in value distributions and mitigates the non-stationarity induced by policy updates. It has been applied in offline RL under distributional shift [CODAC, NeurIPS’21] and in model-based RL to quantify uncertainty [EQR, JMLR’24], but it remains less studied in meta-RL, where expectation-based value estimation is often sufficient for parametric tasks with similar returns. Recently, SDVT and AMAGO-2 consider more practical non-parametric settings with fundamentally different tasks.
>
> AMAGO-2 is, to the best of our knowledge, the first to model the full distribution of meta-values for heterogeneous tasks. However, its discretization-based design relies on predefined bins and value ranges, which can lead to inaccurate estimates when return scales vary significantly. In contrast, RTC leverages Sinkhorn distances for meta-value learning, enabling adaptive modeling across different return scales. Moreover, we introduce a recursive error model to prioritize critical samples and mitigate statistical bias during Sinkhorn updates.
>
> **W3Q1:**
>
> Eq.(8) is not directly tractable due to the unknown true meta-value discrepancy. We consider the tractable relaxation according to the Fenchel-Young inequality, but both have the same minimum $\tilde{v}^*$. Since it is difficult to directly obtain samples from the training distribution $p_k$ via online interaction, we use samples from a buffer distribution $\mu_k$ and reweight these samples using importance sampling. The resulting Eq.(12) is thus theoretically motivated, consistent with the heuristic intuition of down-weighting high-error samples, and proves effective in practice. We will rephrase the relevant description.
>
> **Q3:** Due to word limits, please refer to our first response to Reviewer JfTG.
>
> **Q4:**
>
> Our RTC is robust across seeds and hyperparameters. As illustrated in Table 2, RTC exhibits lower variance than the baselines. Its hyperparameters are fixed across ML-10 and ML-45, and additional experiments indicate low sensitivity to their choices. As shown in Table 7 below, an appropriate regularization coefficient $\lambda$ balances smoothness and accuracy, while increasing the Sinkhorn iteration number $l$ yields more accurate meta-value estimates and higher test success rates.
>
> **Table 7: Converged average test success rate ± standard error (\%) with different $\lambda$ and $l$ values.**
>
> |Task|RTC $(\lambda=1,l=10)$|$\lambda=0.1$|$\lambda=10$|$l=5$|$l=15$|
> |-|-|-|-|-|-|
> |ML-10|30.39±1.62|29.38±2.32|23.41±3.64|27.86±2.29|32.90±1.39|

---

> > ### Author Rebuttal · Reviewer_y6Un · 2026-04-03
> >
> > Thanks for the authors rebuttal

---

> > > ### Author Response · Authors · 2026-04-03
> > >
> > > Dear Reviewer,
> > >
> > > Thank you very much for your feedback. We are glad to know that our rebuttal has fully resolved your concerns. If you could please reflect this in your score as well, we would greatly appreciate it.
> > >
> > > We will revise the paper accordingly. Please let us know if you have any further questions. We would be happy to address them before the end of the discussion period.
> > >
> > > Thank you again for your time and consideration.
> > >
> > > Best regards,
> > >
> > > Authors

---

### Decision · Program_Chairs · 2026-04-30

**Decision:**

Accept (regular)

**Comment:**

This paper introduces Reflect-then-Correct (RTC), a novel meta-RL framework that tackles heterogeneous task returns via Sinkhorn divergence for distributional value alignment and a recursive error model for adaptive correction. It demonstrates strong empirical performance on Meta-World benchmarks.

The key strengths are its innovative methodology and solid experimental results. However, reviewers noted weaknesses in presentation clarity and questioned the method's generalizability beyond Meta-World. The authors' rebuttal successfully addressed most technical concerns, though one reviewer remains unconvinced on broader impact.